

# Understanding the Mid-Pleistocene transition with a simple physical model

Sergio Pérez-Montero[1,2], Jorge Alvarez-Solas[1,2], Jan Swierczek-Jereczek[2], Daniel Moreno-Parada[3], Alexander Robinson[4], and Marisa Montoya[1,2]

[1]Geosciences Institute (IGEO, CSIC-UCM), Madrid, Spain
[2]Complutense University of Madrid (UCM), Department of Earth Physics and Astrophysics, Madrid, Spain
[3]Université Libre de Bruxelles (ULB), Laboratoire de Glaciologie, Brussels, Belgium
[4]Alfred Wegener Institute, Helmholtz Centre for Polar and Marine Research, Potsdam, Germany

**Correspondence:** Sergio Pérez-Montero (sepere07@ucm.es) and Jorge Alvarez-Solas (alvarez.solas@igeo.ucm-csic.es)

**Abstract.** The climate of the Quaternary period is dominated by glacial-interglacial variability due to changes in the Earth's orbital parameters that control the incoming solar radiation. However, certain features of this variability remain puzzling. A notable example is the so-called Mid-Pleistocene Transition (MPT, circa 1 million years ago), characterized by the shift of the predominant periodicity in climate variability from 40 kyr during the Early Pleistocene to 100 kyr at the Late Pleistocene.
Previous studies have tried to explain its origin by invoking two main hypotheses. The first one is based on the observed decreasing trends in temperature and $CO_2$ throughout various climatic proxies. The second one, the regolith hypothesis, is based on the change in the basal friction regime of the Northern Hemisphere ice sheets via a progressive elimination of sediment layers of sediments above the continents. Here, we use the Physical Adimensional Climate Cryosphere mOdel (PACCO) to reproduce orbital-scale climate variability throughout the entire Pleistocene through a physical albeit simplified approach. We
find that the decreasing trends in $CO_2$ and temperature during the Pleistocene can be explained with PACCO as a consequence of an MPT triggered by regolith removal that changes the size of the Northern Hemisphere ice sheets. The pre- and post MPT world respectively yield dominant periodicities around 40 and 100 kyr, the timing of the MPT corresponds to what is observed in proxies and the amplitude of sea-level changes is well matched.

## 1 Introduction

The Pleistocene climate is governed by glacial-interglacial variability (GIV, Esmark, 1824, 1826; Berger, 1988; Paillard,
2001, 2015; Berends et al., 2021a; Ganopolski, 2024), which is characterized by quasi-periodic oscillations of the global ice volume resulting from the nonlinear response of the climate system to changes in the incoming solar radiation (Agassiz, 1840; Adhémar, 1842; Murphy, 1876; Milankovitch, 1941; Paillard, 2001; Chalk et al., 2017; Ganopolski, 2024, Fig. 1). These cycles exhibit a sawtooth pattern (Broecker and Van Donk, 1970) which consists of gradual glaciations and abrupt deglaciations
(Fig. 1). Proxy records indicate that, around 1250-700 kyr before present (BP), the characteristic climate variability underwent a transition from 40 kyr to 100 kyr cycles (Clark et al., 2006). This is known as the Mid-Pleistocene Transition (MPT, Chalk




et al., 2017). Furthermore, the amplitude of climate variability during the Early Pleistocene is significantly smaller than that of the Late Pleistocene.

The physical mechanisms behind the MPT are not fully identified. In the past decades various mechanisms invoking ice
sheets, sea ice, the ocean circulation and the carbon cycle have been proposed to explain this transition (for a complete review see Berends et al., 2021a). The main hypotheses to date are the regolith hypothesis (Pisias and Moore Jr, 1981; Berger and Jansen, 1994; Clark and Pollard, 1998; Clark et al., 2006; Ganopolski and Calov, 2011; Willeit et al., 2019; Carrillo et al., 2024) and the cooling trend in climate conditions (due to the gradual decrease in the atmospheric $CO_2$ concentration, Raymo et al., 1997; Tziperman and Gildor, 2003; Paillard and Parrenin, 2004; Hodell and Venz-Curtis, 2006; Clark et al., 2006; Verbitsky
et al., 2018; Willeit et al., 2019; Carrillo et al., 2024; Clark et al., 2024). The reader is referred to Chalk et al. (2017), Berends et al. (2021a) and Martin et al. (2024) for extensive reviews on this matter. The regolith hypothesis is based on the existence, at the beginning of the Pleistocene, of a thick layer of unconsolidated sediments (regoliths) formed during the warmer Pliocene epoch (5.33 to 2.58 million years BP) that used to cover the continents and that could have affected the dynamics of the ice sheets through enhanced sliding (Fig. 2, Clark and Pollard, 1998; Willeit et al., 2019; Ganopolski, 2024). In this way, at the
Early Pleistocene, ice-sheet flow would have been faster, making ice sheets more sensitive to surface mass balance changes produced by relatively small anomalies in incoming radiation. However, the basal ice flow gradually advected the sediments outside of the ice sheets, thus thinning the sediment layer and reducing basal velocity. The overall ice flow therefore became slower and less insolation-sensitive, and other processes started to gain importance as triggers of glacial terminations (for a full description of the hypothesis, see Clark et al., 2006). The cooling trend hypotheses are employed by most of the models
and are supported to some extent by proxy evidence (and model-assisted proxy evidence, Berends et al., 2021a) that imply a decreasing atmospheric temperature and $CO_2$ concentration across the Pleistocene due to reduced volcanic outgassing or increased weathering (Clark et al., 2006; Willeit et al., 2019; Berends et al., 2021b). This trend toward a colder climate would in principle facilitate the growing of bigger ice sheets, that could have lasted longer in the Late Pleistocene.

Pérez-Montero et al. (2024) described the new conceptual model PACCO (Physical Adimensional Climate Cryosphere
mOdel) and showed its capability to reproduce the Late-Pleistocene GIV. Experiments with PACCO showed that ice-sheet dynamics, glacial isostatic adjustment and the delayed effect of the ice-sheet thermodynamics on basal sliding lead to slow glaciations and fast glacial terminations of the appropriate amplitude. Together with the darkening of ice through enhanced dust deposition with time, the model produced 100-kyr glacial cycles in agreement with proxies and reconstructions both in timing and in shape. In this work, we will use PACCO to explore the effect of ice velocity on the sensitivity of the Northern
Hemisphere ice sheets to climate boundary conditions and its role in relation with the MPT. Thus, we will focus on the regolith hypothesis. To this end, PACCO is extended to represent sediment layer dynamics. In Sect. 2 we briefly describe the model employed and the new processes that are included. In Sect. 3 we present the main results of this work. In Sect. 4 we discuss them. Finally, in Sect. 5, we summarize our main conclusions.





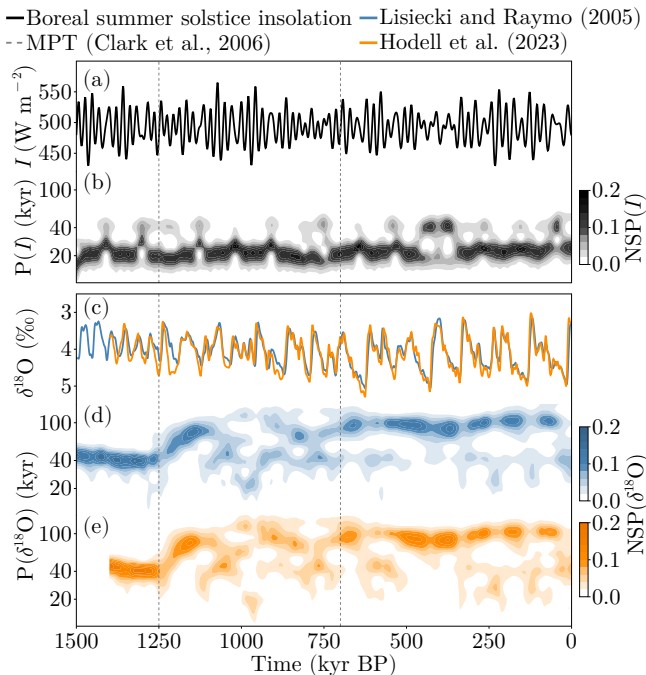

**Figure 1.** (a) Boreal summer solstice insolation (SSI) at 65°N calculated following Berger (1978) and (b) its wavelet transform. (c) Lisiecki and Raymo (2005) and Hodell et al. (2023) $\delta^{18}O$ time series for the last 1.5 million years and (d, e) their respective wavelet transforms. Note that wavelet transforms in (b, d, e) show the normalized (integrated) spectral power (NSP) at each period ($P$) as a function of time. Vertical dashed lines correspond to the interval associated with the occurrence of the MPT (Clark et al., 2006). Proxy records were filtered with a lowpass Butterworth filter (cutoff frequency of 10 kyr$^{-1}$).

## 2   Model description

PACCO, as introduced by Pérez-Montero et al. (2024), represents the interaction between the Northern Hemisphere ice sheets and climate by eliminating the spatial dimensions of ice-sheet dynamics equations and coupling them to a simple climate model that translates insolation forcing ($I$, only boundary condition) to ice-sheet mass balance. Thus, the prognostic variables of the model are: regional air temperature ($T$, Appendix A1), CO$_2$ concentration ($C$, Appendix A2), ice thickness ($H$, Appendix A3) and bedrock elevation ($B$). In the present work, we further include the sediment layer thickness ($H_\mathrm{sed}$, Sect. 2.1), following the

scheme in Fig. 3. We describe the governing equation of the sediment layer thickness below and refer to Pérez-Montero et al. (2024) for all processes that are not described here.





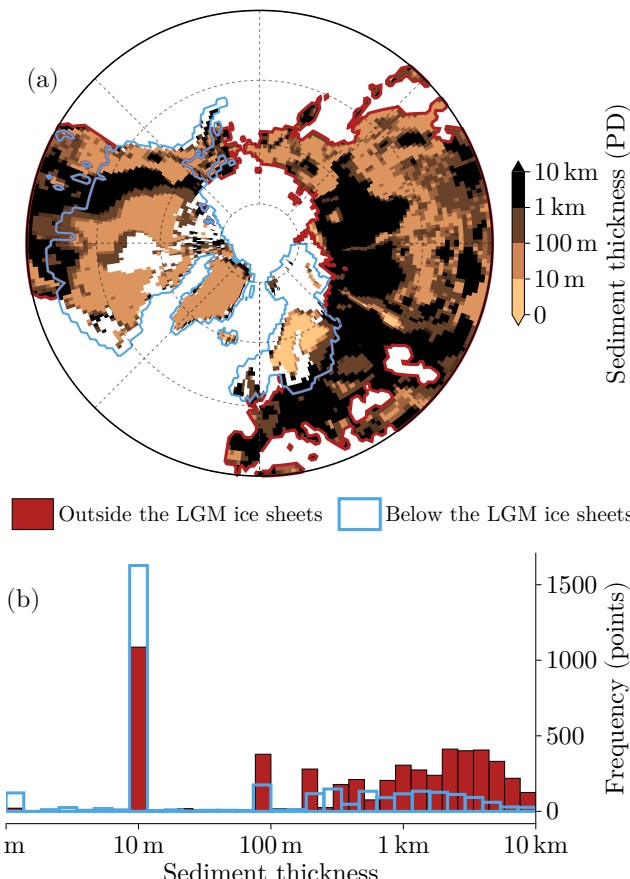

**Figure 2.** (a) Present-day (PD) sediment spatial distribution above PD sea level from Mooney et al. (2023) together with ice sheet extent during Last Glacial Maximum (LGM, blue colors) from Roy and Peltier (2017, ICE7G) and (red colors) PD emerged continents outside the maximum extent of the LGM ice sheets. (b) Histograms of sediment thickness in the regions delineated in (a).

## 2.1 Adding sediment layer dynamics to PACCO

In Pérez-Montero et al. (2024), ice-sheet dynamics are represented by a velocity calculation that accounts for both deformational and sliding regimes in an ice sheet:

$$v = \frac{2}{5} \cdot A_f \cdot H \cdot \tau_d^3 + \beta \cdot f_{\text{str}} \cdot C_s \cdot \tau_b^2, \tag{1}$$

where

$$\tau_d = \frac{\rho_{\text{ice}} \cdot g}{c} \cdot \frac{H}{z} \tag{2}$$




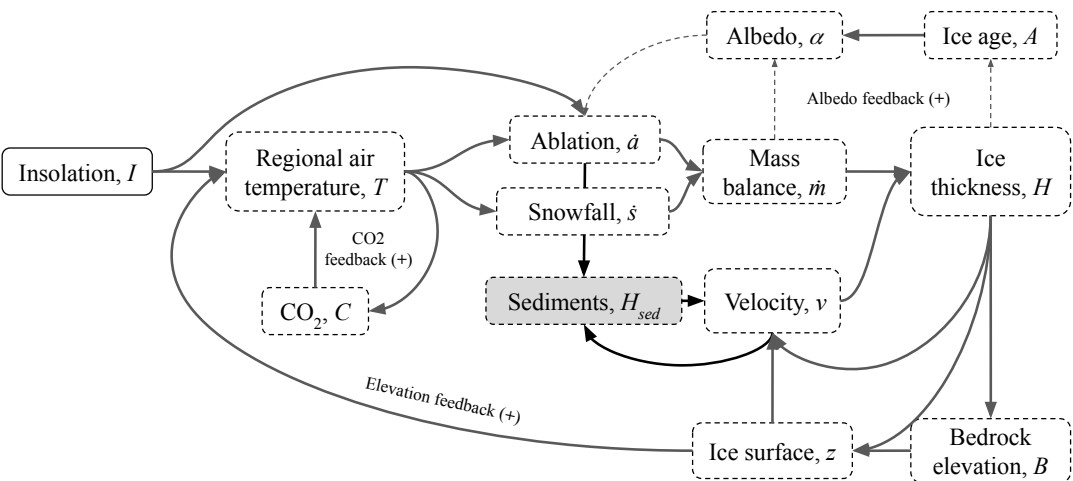

**Figure 3.** PACCO processes flowchart. Note that we highlight in black the new process added to the model and its interaction with the rest of the components.

**Table 1.** Model parameters.

| Parameter | Name | Reference | Value (range) | Units |
|---|---|---|---|---|
| $A_f$ | Glen's law flow parameter | Glen (1958) | $10^{-16}$ | $\mathrm{Pa}^{-3}\,\mathrm{yr}^{-1}$ |
| $c$ | Ice surface profile coefficient | Verbitsky et al. (2018) | 0.9 | $\mathrm{m}^{-1}$ |
| $C$ | Constant $CO_2$ level applied | | 100-380 | ppm |
| $C_s$ | Sliding coefficient | Pollard and DeConto (2012) | $6 \cdot 10^{-5}$ | $\mathrm{m}\,\mathrm{yr}^{-1}\,\mathrm{Pa}^{-2}$ |
| $g$ | Gravitational acceleration | | 9.81 | $\mathrm{m}\,\mathrm{s}^{-2}$ |
| $f_{\dot{p}}$ | Denudation rate fraction | Clark and Pollard (1998); Cuffey and Paterson (2010) | $3 \cdot 10^{-5}$ ($10^{-7}$-$10^{-3}$) | |
| $f_{\dot{v}}$ | Sediment flux fraction | Jamieson et al. (2008); Golledge et al. (2013); Cook et al. (2020) | $1.4 \cdot 10^{-7}$ ($5 \cdot 10^{-8}$-$10^{-6}$) | |
| $f_{\mathrm{str}}$ | Fraction of ice streams in the ice sheet | Margold et al. (2015) | 0.2 | |
| $H_{\mathrm{sed,max}}$ | Maximum amount of sediments allowed | Clark et al. (2006) | 30 | m |
| $H_{\mathrm{sed,min}}$ | Minimum amount of sediments allowed | Based on Fig. 2b | 5 | m |
| $\rho_{\mathrm{ice}}$ | Glacier ice density | | 910 | $\mathrm{kg}\,\mathrm{m}^{-3}$ |
| $\theta$ | Ice stream profile scale | Benn et al. (2019) | $10^{-3}$ | $\mathrm{K}\,\mathrm{m}^{-1}$ |

is the driving stress under which gravity deforms the ice, and

$$\tau_b = \rho_{\mathrm{ice}} \cdot g \cdot H \cdot \theta \qquad (3)$$

is the basal stress that determines the ice-sheet sliding over the underlying bedrock. In Eqs. (1), (2) and (3) $A_f$ is the Glen's law flow parameter that accounts for ice viscosity (Glen, 1958), $f_{\mathrm{str}}$ is the fraction of ice streams in the ice sheet (Margold et al., 2015), $C_s$ is a sliding parameter derived from Pollard and DeConto (2012), $\rho_{\mathrm{ice}}$ is the ice density, $g$ is the gravitational acceleration, $c$ is the amplitude of the assumed parabolic profile for determining ice deformation (following Verbitsky et al.,





2018, see Pérez-Montero et al., 2024) and $\theta$ is the slope of the ice-stream region (Benn et al., 2019). $H$ is the ice-sheet thickness, whose evolution is defined in Appendix A3. $\beta$ is the novelty in the model and it is a fraction between 0 and 1 that represents the potential sliding of the Northern Hemisphere ice sheets and is defined as:

$$\beta = \frac{H_{\text{sed}} - H_{\text{sed,min}}}{H_{\text{sed,max}} - H_{\text{sed,min}}}, \tag{4}$$

where $H_{\text{sed,min}}$, $H_{\text{sed,max}}$ are paleoclimatic constraints on minimum (e.g. present day, see Fig. 2) and maximum (e.g. Pliocene, Goldich, 1938; Setterholm and Morey, 1995; Clark et al., 2006) values of sediment layer thickness. $\beta$ represents the anomaly of sediment coverage with respect to the early Pleistocene and therefore captures how likely it is for the ice sheet to reach sediment-covered areas (based on sediment masks from Ganopolski and Calov, 2011; Willeit et al., 2019). Of course, the choice of the function shaping $\beta$ is arbitrary; we selected a linear relationship for the sake of simplicity. However, the versatility of PACCO allows the implementation of more sophisticated relationships. Finally, $H_{\text{sed}}$ evolves in time following:

$$\frac{dH_{\text{sed}}}{dt} = f_{\dot{p}} \cdot \dot{p} - f_v \cdot v. \tag{5}$$

The first term on the right hand side (RHS) represents the denudation rate, which is the process that increases the presence of sediments via weathering (Clark and Pollard, 1998; Cuffey and Paterson, 2010). Thus, $f_{\dot{p}}$ is the fraction of precipitation $\dot{p}$ that produces new sediments (Lebedeva et al., 2010). And $\dot{p}$ is equal to snowfall $\dot{s}$ when there is no ice (i.e. $H = 0$, only during interglacials). This choice is motivated by the assumption that during interglacials the climate is warmer and accumulation is given by liquid water instead of snow, thus eroding the rock instead of accumulating snow. The second RHS term is the sediment flux produced by the quarrying of the bed due to the movement of the ice sheet (Jamieson et al., 2008; Golledge et al., 2013; Cook et al., 2020). $f_v$ then represents the proportion of sediments that are eliminated because of the ice-sheet flow. Combining Eqs. (1), (4) and (5), it can be seen that the evolution of $H_{\text{sed}}$ translates into a change in the sliding of the ice sheet, thus changing its total velocity.

Finally, $z$ is the ice surface elevation that is defined as

$$z = H + H_{\text{sed}} + B. \tag{6}$$

Including interactive sediments in PACCO results in an additional negative feedback between sediment thickness and ice velocity (Fig. 3), which directly affects the dynamics (Eq. 1) and mass conservation of the ice sheet (Eq. A4). The Appendix A is dedicated to further selected equations, already introduced by Pérez-Montero et al. (2024), which are relevant in the context of the present study.

# 3 Results

In this section we describe the experiments performed and analyze their main results. To do so, we first show a baseline experiment with the standard model setup that reproduces the entire Pleistocene (Sect. 3.1). Then, we test the sensitivity of PACCO to the parameters of Eq. (5) in Sect. 3.2. Next, an experiment to assess the amplifying role of $CO_2$ is carried out in





**Table 2.** Summary of the experiments performed in this work.

| Experiment | Description | Section |
|---|---|---|
| BASE | Simulation of the last 3 Myr using the 65°N 21 June insolation | 3.1 |
| SEDIM | Sensitivity experiment applied to BASE to explore the effect of the denudation rate $f_{\dot{p}}$ and the sediment flux $f_v$ | 3.2 |
| CARBON | Sensitivity experiment to test the role of $CO_2$ when applying constant levels of carbon dioxide to BASE | 3.3 |
| INSOL | Sensitivity experiment to see the effect of using different insolation metrics in the model's response | 3.4 |
| NOSEDIM | Simulation of the last 3 Myr with deactivated dynamic sediments for two levels of sliding sensitivity (high and low) | 3.5 |
| TREND-C | Sensitivity experiment using NOSEDIM-low configuration and imposing a trend in the carbon cycle | 3.5.1 |
| TREND-S | Sensitivity experiment using the NOSEDIM-low setting and imposing a trend in accumulation sensitivity to temperature | 3.5.2 |

Sect. 3.3, then, we investigate the definition of insolation forcing and its effect on the pre-MPT response in Sect 3.4, and finally, we test other hypotheses beyon the regoliths in Sect. 3.5. All experiments are summarized in Table 2.

## 3.1 BASE experiment

The BASE experiment (Table 2) is a simulation of the entire Quaternary (from 3 Myr BP to present) obtained by forcing PACCO with the boreal summer solstice insolation (SSI) at 65°N following Berger (1978). The values of the parameters and initial conditions can be found in Tables 1 and A1. We use a model configuration that includes ice-sheet dynamics, isostasy, carbon cycle and albedo darkening (i.e. the AGING configuration in Pérez-Montero et al., 2024), and by including the basal sliding dependence on interactive sediment evolution we obtain good qualitative agreement between simulation results and proxies, including a change in the periodicity of the system around 1250-700 kyr BP, the MPT.

As can be seen in Fig. 4, the GIV in BASE is generally in good agreement with proxies, especially the sea-level evolution over the last 800 kyr. Most importantly, it displays a change in both amplitude and periodicity at the MPT. This change is caused by the gradual decrease of the sediment thickness (second term of the RHS of Eq. 5) that decreases the ice velocity (Eqs. 1 and 4) and ultimately increases the ice thickness through Eq. (A4) (Fig. 5). Before the MPT, the ice-volume cycles are faster and smaller since the sediment thickness is higher (Fig. 4f) and the ice-sheet flow is generally faster (Figs. 5b and 6c). Thus, the ice sheet is more sensitive to precession and obliquity periods, which leads to smaller circular trajectories when projecting the phase space onto the insolation and ice-thickness dimensions (Fig. 6b). After the MPT, the gradual reduction in the sediment thickness results in slower ice sheets (Fig. 6c). Therefore, ice sheets can grow bigger, which allows them to



**Figure 4.** Results of BASE experiment: (a) Boreal SSI, (b) regional air temperature anomaly with respect to $T_{\mathrm{ref}}$ in comparison with Bintanja and Van de Wal (2008) and Barker et al. (2011), (c) CO₂ concentration compared with Lüthi et al. (2008) and Berends et al. (2021b), (d) ice thickness, (e) sediments thickness and (f) ice volume time series in comparison with Bintanja and Van de Wal (2008) and Spratt and Lisiecki (2016) and superimposed to $\delta^{18}O$ from Lisiecki and Raymo (2005). Note that black curves are calculated by PACCO and the coloured lines correspond to the proxies and reconstruction of the legend.



persist over several obliquity cycles. As a consequence, glacial cycles last longer compared to the Early Pleistocene (Figs. 5c and 6b). This is reflected in the wavelet transform of the simulated and reconstructed ice volume (Fig. 7). The power spectrum is concentrated around 20-40 kyr before the MPT and around 80-120 kyr after. Hence, compared with proxy records, lower pre-MPT periodicities are obtained. This could be due to the fact that the SSI we use shows a high precessional spectral power (Leloup and Paillard, 2022).

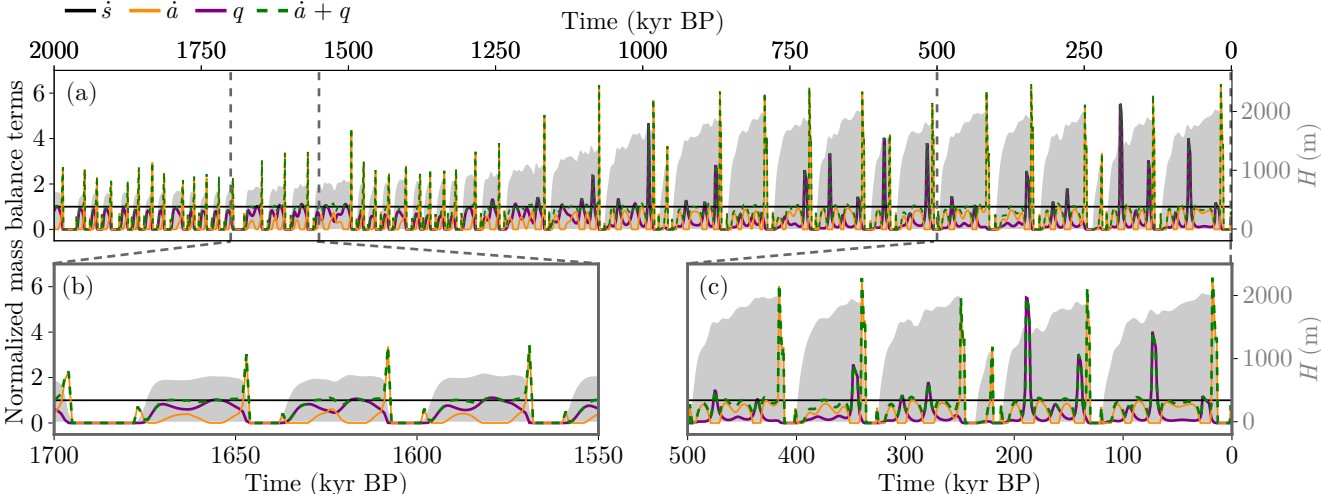

**Figure 5.** Results from BASE experiment showing the evolution of the ice thickness $H$ (in gray) and mass balance terms (snowfall $\dot{s}$, ablation $\dot{a}$ and ice discharge $q$ normalized by $\dot{s}$, as defined in Appendix A3) for different ranges of time.


As introduced in Sect. 1, the MPT has been suggested to occur because of the decreasing trend in $CO_2$ or temperature. In our model introducing ad-hoc temperature or $CO_2$ trends is not needed since they are produced naturally by the model via the cooling effect of the gradually thicker ice sheets throughout the Pleistocene ($c_z \cdot z$ in Eq. A1). The modulation of the ice-surface elevation $z$ via basal dynamics provides us with such trends in climate. Thus, these trends in PACCO are consequences rather

than triggers of the MPT.

As we described above, the gradual reduction of the sediment thickness plays a key role in producing an MPT that is coherent with proxy records. We therefore propose to study the sensitivity of the results to the parameters controlling the evolution of $H_{\mathrm{sed}}$.

### 3.2 Sensitivity to sediment evolution (SEDIM experiment)

In Eq. (5) we use two parameters, the denudation and quarrying fractions ($f_{\dot{p}}$ and $f_v$, respectively) to represent a rather complex process in a simple manner. Thus, we perform a sensitivity experiment (SEDIM, Table 2) to investigate the effect of the values of these parameters selected in BASE. Golledge et al. (2013) propose $10^{-6}$ m yr$^{-1}$ as a reasonable value for the quarrying fraction $f_v$. They found values of sediment fluxes ranging from 100 to more than 1000 m$^3$ yr$^{-1}$, which corresponds





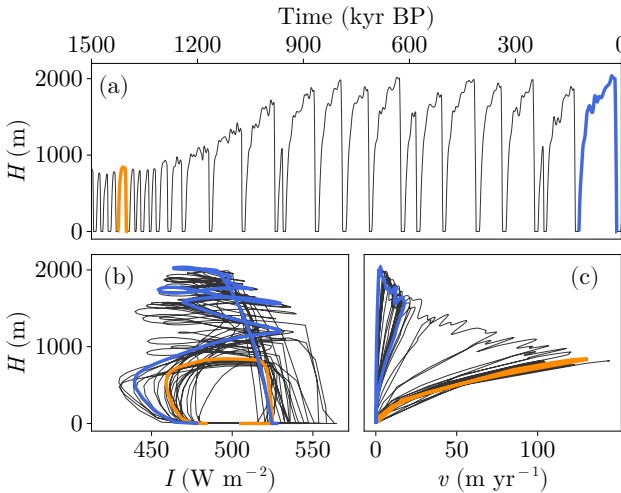

**Figure 6.** (a) Ice thickness time series from BASE experiment. (b, c) Ice thickness trajectories from BASE experiment as a function of insolation forcing and ice sheet velocity.

to $f_v \in [10^{-8}, 10^{-6}]$ for a typical ice sheet area ($L^2$) of about $10^{10}$ to $10^{12}$ m$^2$. Moreover, the denudation fraction $f_{\dot{p}}$ is
estimated to be about $10^{-6}-10^{-5}$ m yr$^{-1}$ in west Greenland (Strunk et al., 2017) during the last 1 million years, $1.5 \cdot 10^{-5}$ m yr$^{-1}$ in the Svalbard archipelago during the Holocene (Svendsen et al., 1989), and up to $10^{-3}$ m yr$^{-1}$ in the Alps (Delunel et al., 2020). These two parameters control the time when the MPT occurs, as shown in Fig. 8, defined as

$$t_{\mathrm{MPT}} = t(H_{\mathrm{sed}} = 6), \tag{7}$$

since we empirically observed a threshold in the oscillatory regime around that value of sediment thickness. Figure 8 shows
two regimes: If $f_{\dot{p}}$ is below $10^{-5}$, $f_v$ dictates the timing of MPT (ice dynamic regime). Above $10^{-5}$, $f_{\dot{p}}$ becomes important (precipitation regime). This highlights the fact that, as long as the weathering fraction partially equilibrates the quarrying fraction, the run transitions from one regime to the other with the right timing (i.e. during the MPT period observed in the proxies, Clark et al., 2006). This is represented by Fig. 9, which shows the evolution of $H$ and $H_{\mathrm{sed}}$ fixing one parameter and changing the other one around the BASE case. Depending on the value of $f_{\dot{p}}$, interglacials generate more or less sediments to
the point of overweighting sediment removal. Something similar occurs when fixing $f_{\dot{p}}$, but in this case, if $f_v \leq f_{\dot{p}}$, we can achieve the right timing of the MPT. Finally, as shown in Fig. 8, BASE displays a correct timing of the MPT which can however be also obtained with other parameter combinations. Of course, the particular calibration of the run can in principle alter Fig. 8. However, for a fixed parameter space, the selection of $f_{\dot{p}}$ and $f_v$ depends on the partial balance between both contributions, which results in the regimes described above (ice and precipitation regimes).



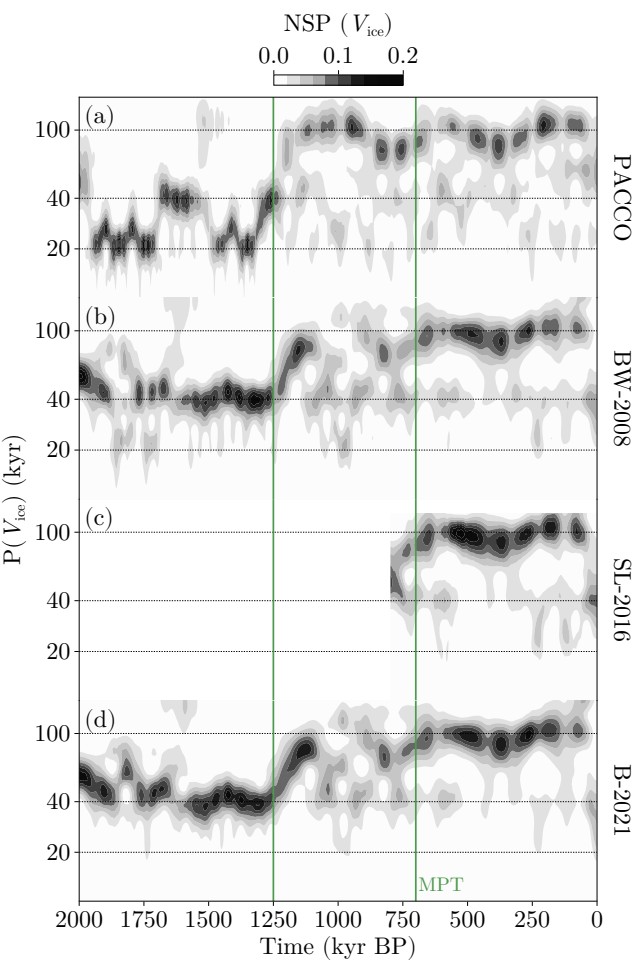

**Figure 7.** NSP of ice volume $V_{ice}$ at each period ($P$) as a function of time from (a) BASE experiment, (b) Bintanja and Van de Wal (2008, BW-2008), (c) Spratt and Lisiecki (2016, SL-2016) and (d) Berends et al. (2021b, B-2021). Vertical green lines correspond to the interval associated with the occurrence of the MPT (Clark et al., 2006).



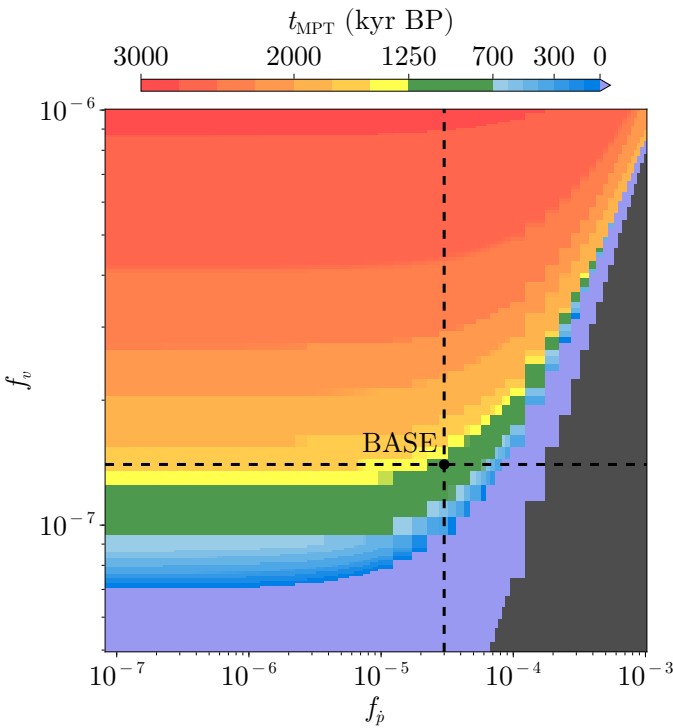

**Figure 8.** MPT onset ($t_{MPT}$) for an ensemble of 9240 evenly spaced permutations of $f_v$ and $f_{\dot{p}}$ (Note that the BASE experiment is represented by the big black dot). Green colors show transitions that start between 1250 and 700 kyr BP (Clark et al., 2006) and the gray zone represents the runs in which the sediment thickness shows an increasing rather than a decreasing trend. Finally, black dashed lines correspond to the curves shown in Fig. 9.

### 3.3 The amplifying effect of CO$_2$ (CARBON experiment)

Previous work (e.g., Willeit et al., 2019) has analyzed the role of CO$_2$ in the GIV, concluding that its variability is not as important as its average background value. To verify this in PACCO, we turn off the carbon cycle component and perform a series of experiments where we apply a constant level of atmospheric CO$_2$. Otherwise, the rest of the parameters are the same as in BASE. The results for several atmospheric CO$_2$ levels are presented in Fig. 10. In PACCO, CO$_2$ shows the same amplifying role as in other studies (in agreement with Ganopolski and Calov, 2011; Willeit et al., 2019). Since CO$_2$ affects the temperature, the accumulation changes and the cycles decrease their amplitudes depending on the level of this gas. During the pre-MPT world, the median of the temperature distribution in each run increases with the imposed background atmospheric CO$_2$ level (Fig. 10h-l). However, the median temperature of the post-MPT world is nearly the same except for the highest atmospheric CO$_2$ level inspected, 380 ppm. The climate in this run is especially warm. Accordingly, ablation is greatly favored, making the denudation term in Eq. (5) greater than the quarrying one for the entire Pleistocene. Thus, the transition does not occur (Fig. 10b). For atmospheric CO$_2$ levels in the range of 100-300 ppm, the model always transitions to the post-MPT world. Runs

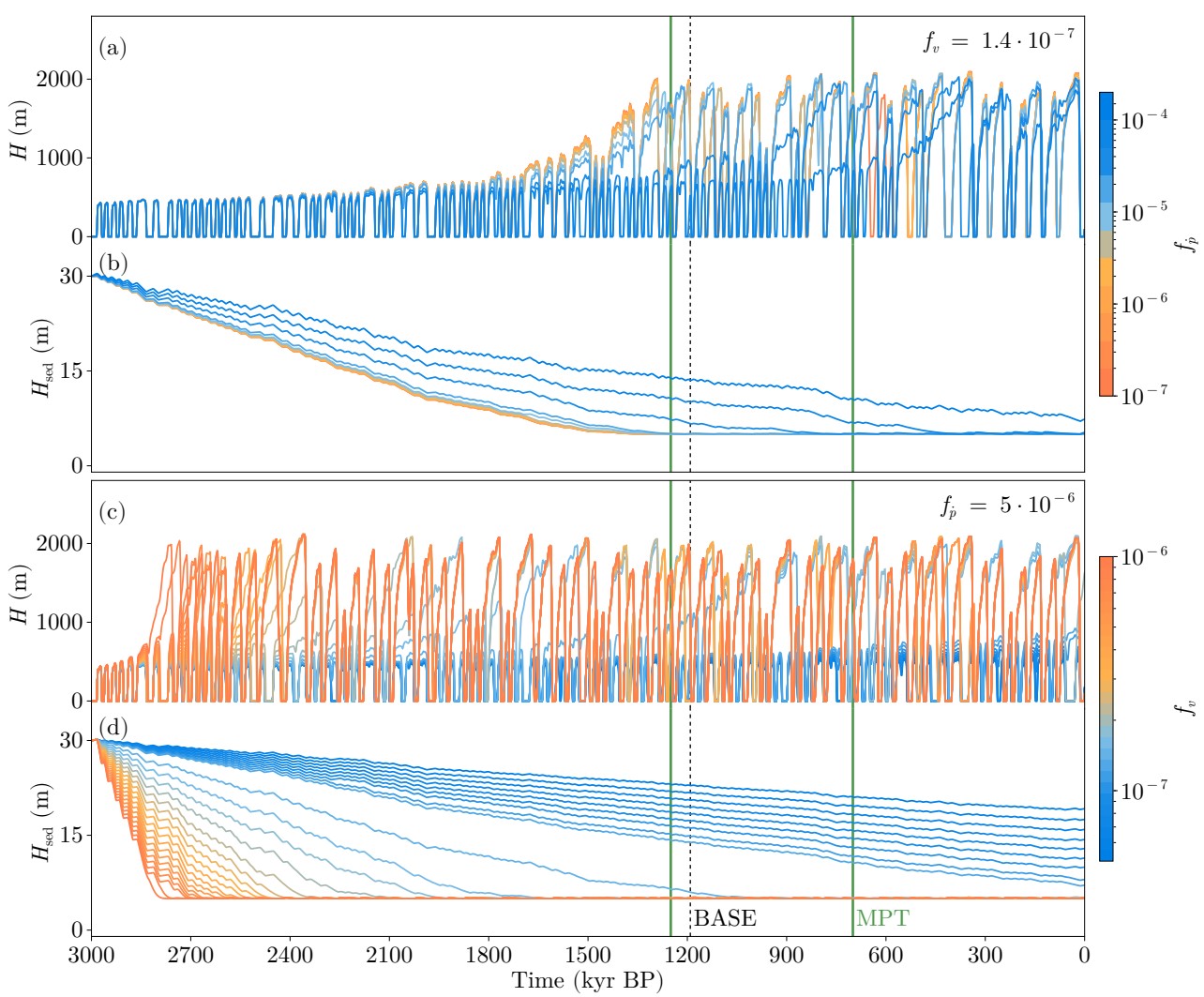

**Figure 9.** Time series for two experiments varying $f_v$ and $f_{\dot{p}}$. Note that the vertical green lines indicate the bounds of the MPT as found in the proxies (Clark et al., 2006) and the dashed black and vertical line the MPT as obtained in the BASE experiment. Note that we have removed the series whose sediments increase rather than decrease throughout the entire series since their parameter space is not coherent with the paleoclimatic constraints. For clarity we have only plotted every 5th simulation of the ensemble.





using atmospheric $CO_2$ levels between 200-300 ppm are similar to BASE. In contrast, the 100 ppm runs show longer cycles before the MPT. The climate is cold enough to largely inhibit thermal ablation and allow for larger pre-MPT ice sheets that do not experience deglaciation at each obliquity cycle.

In summary, this experiment shows that the interactive carbon cycle in PACCO is an amplifier of GIV rather than the trigger of MPT, in accordance with previous literature results. This finding relies on the fact that the change in regime in PACCO is due to purely dynamic reasons. In our case, the MPT does not need a trend in climate (via $CO_2$ or temperature) since it is produced naturally by the effect of ice sheets on climate.

### 3.4    Insolation forcing definition (INSOL experiment)

In Sect. 3.1 we have seen that the pre-MPT world in PACCO has too much spectral power in the 20 kyr band to be fully coherent with proxy records. In this section we will investigate the effect of the definition of the insolation forcing in this result. Leloup and Paillard (2022) compared the three common definitions of insolation forcing applied in conceptual modeling oriented to orbital-scale variability studies and discussed the relative spectral power of each orbital parameter and their effect on glacier ablation; for a complete description of these metrics see Leloup and Paillard (2022), Tzedakis et al. (2017), Huybers (2006) and

Milankovitch (1941). We define the summer solstice insolation (SSI), the integrated summer insolation (ISI) and the caloric season insolation (CSI) following the previous cited studies as:

$$\text{SSI}(t) = I(t, 170, 65), \tag{8}$$

$$\text{ISI}(t, I_{\text{thr}}) = \sum_{i=1}^{l} w_i \cdot I(t, i, 65), \tag{9}$$


$$\text{CSI}(t) = \sum_{i=1}^{l} w_i^{'} \cdot I(t, i, 65). \tag{10}$$

$I = (t, d, \phi)$ depends on the time $(t)$, the day $(d)$ and on the latitude $(\phi)$ in which we compute insolation following Berger (1978). The weight $w_i$ yields 1 if $I(t, i, 65) \geq I_{\text{thr}}$ and 0 otherwise. In contrast, the weight $w_i^{'}$ yields 1 if $I(t, i, 65)$ is above the median of the year's distribution and 0 otherwise. ISI and CSI are therefore computed as weighted integration of $I$, with $l$ the

summation index that represents the length of the year in days. Note that SSI has units of W m$^{-2}$ and ISI and CSI of J m$^{-2}$ (Fig. B1).

We performed three experiments: BASE (that uses SSI), CSI and ISIn (n indicates the insolation threshold applied to ISI metric). The results are plotted in Fig. 11 using insolation thresholds of 0, 275, 300 and 400 W m$^{-2}$ (i.e., ISI0, ISI275, ISI300 and ISI400, respectively). These selections were made following Huybers (2006, ISI275) and Leloup and Paillard (2022, ISI300 and ISI400).

ISI0 was performed as a contrasting case in which we included the full annual integrated insolation.

After calibrating the insolation sensitivities ($c_I$ and $k_I$ in Eqs. A1 and A7) to account for the fact that in PACCO insolation is expressed as energy and not power, we obtain similar curves to the BASE run. This is particularly the case for the post-MPT

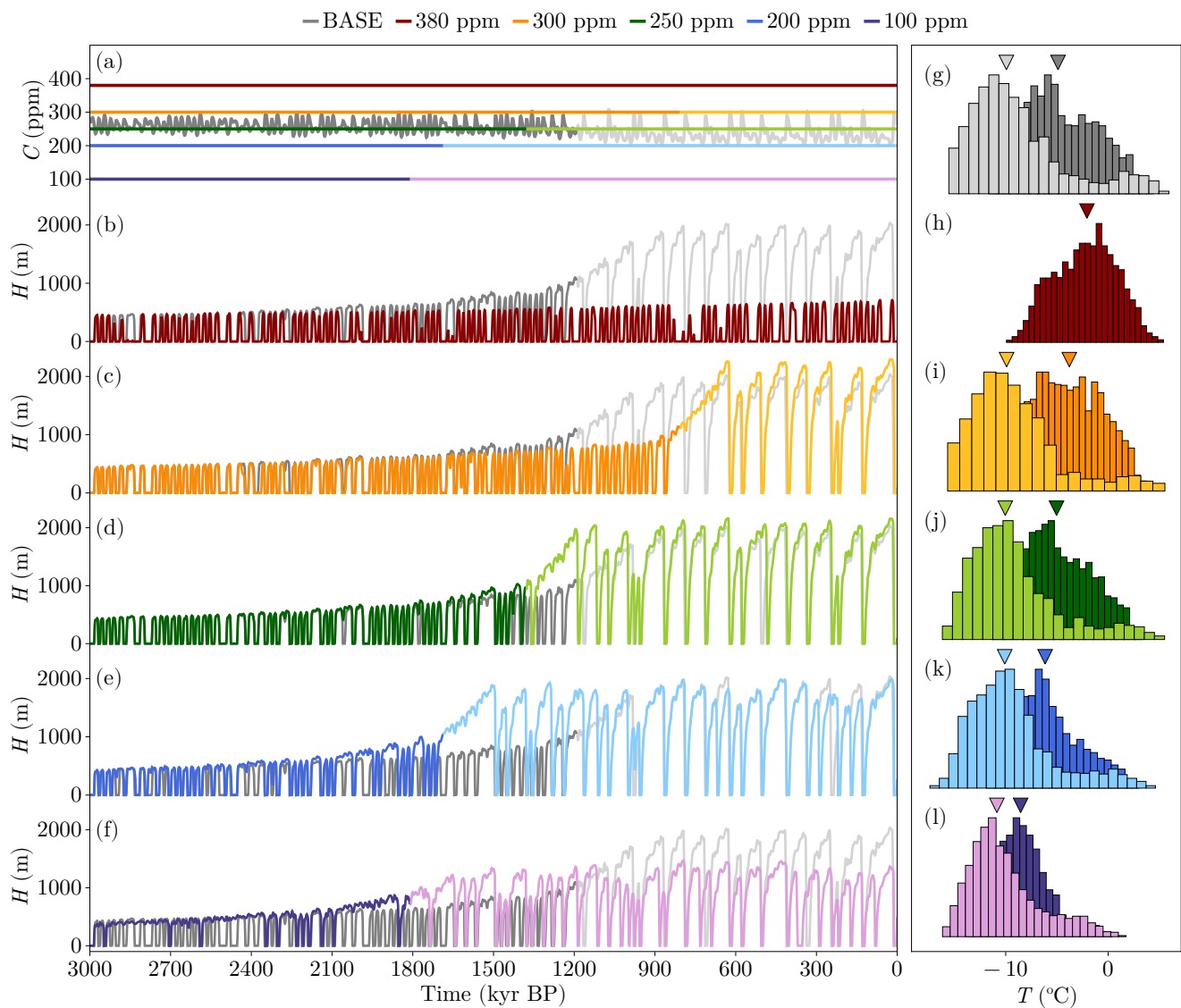

**Figure 10.** (a) Time series of $C$ for each run. Note that darker colors indicate the pre-MPT world ($H_{\mathrm{sed}} > 6$ m) and lighter colors in the post-MPT world. (b-f) Time series of $H$ for each CARBON run and BASE as a reference. (g-l) Histograms (normalized as probability density functions) of $T$ for each CARBON run and BASE. Note that the inverted triangles point to the median of each distribution.





world, as shown in Fig. 11. During the late Pleistocene all series share a similar power spectral density (Fig. 12c), indicating that the particular definition of forcing is not a decisive factor for the post-MPT world. However, in the pre-MPT world, the

spectral power in the CSI and ISIn experiments is located around 40 kyr. This finding is related to the spectral power of each Milankovitch period in these metrics (Fig. 12a). CSI and ISI metrics yield more 40-kyr power since precessional power is eliminated in favor of obliquity when integrating in time (Leloup and Paillard, 2022). In addition, these metrics indirectly alter the capability of the ice sheet to remove sediments since the pre-MPT cycles are longer (compared to BASE) and thus eliminate sediments quickly. For this reason, MPT occurs earlier in CSI and ISIn (Fig. 11).

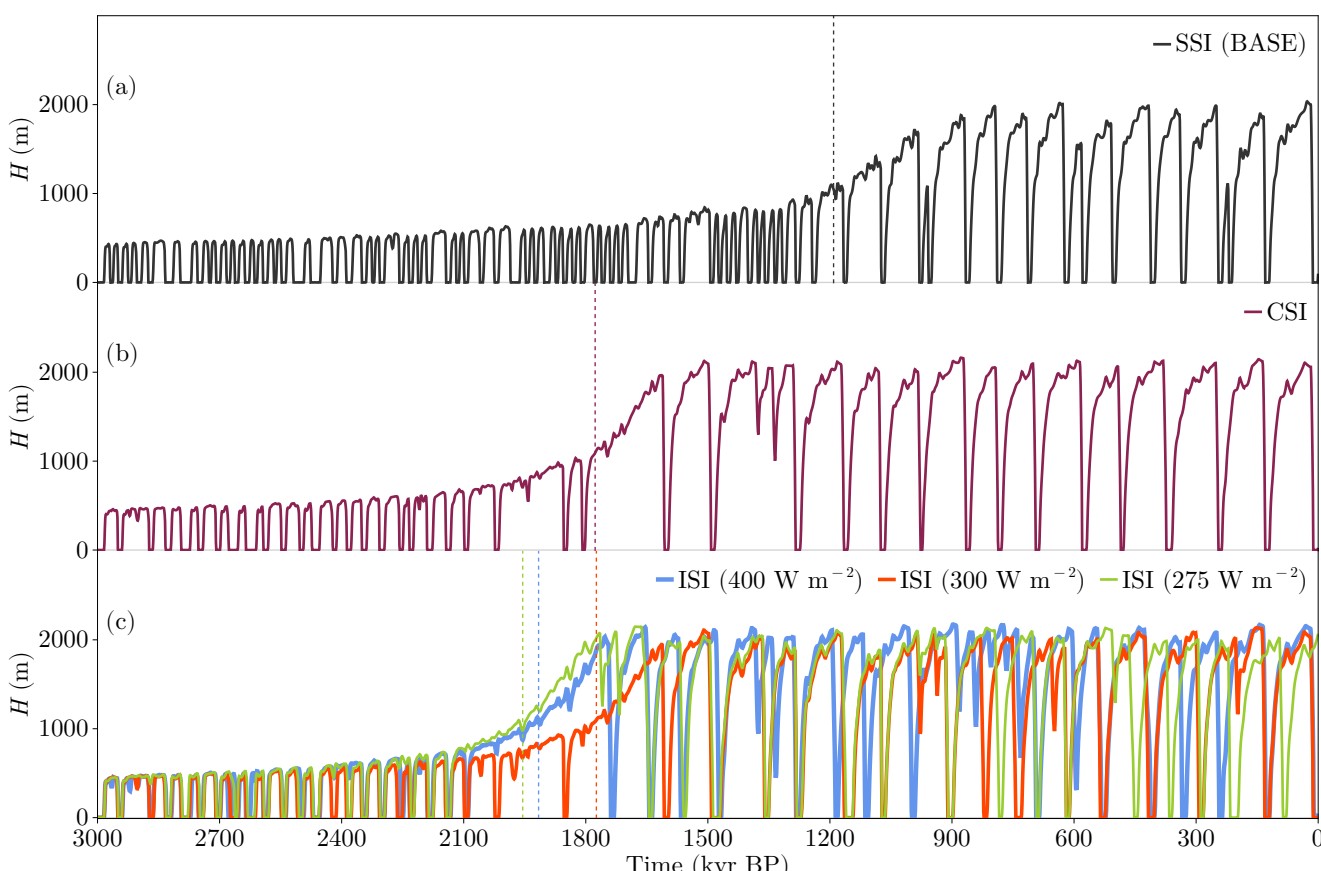

**Figure 11.** (a-d) Time series of ice sheet volume for the different insolation metrics. The particular parameters changed for these runs are located in Table 1. Vertical dashed lines indicate the MPT as defined in the previous section for each run.

Therefore, we conclude that there is an improvement in the pre-MPT world when using CSI or ISI metrics. This improvement does not alter our main finding on the important role of regolith quarrying to trigger the MPT, in contrast with decreasing trends in temperature and $CO_2$, which are consequences of changes in the cryosphere.

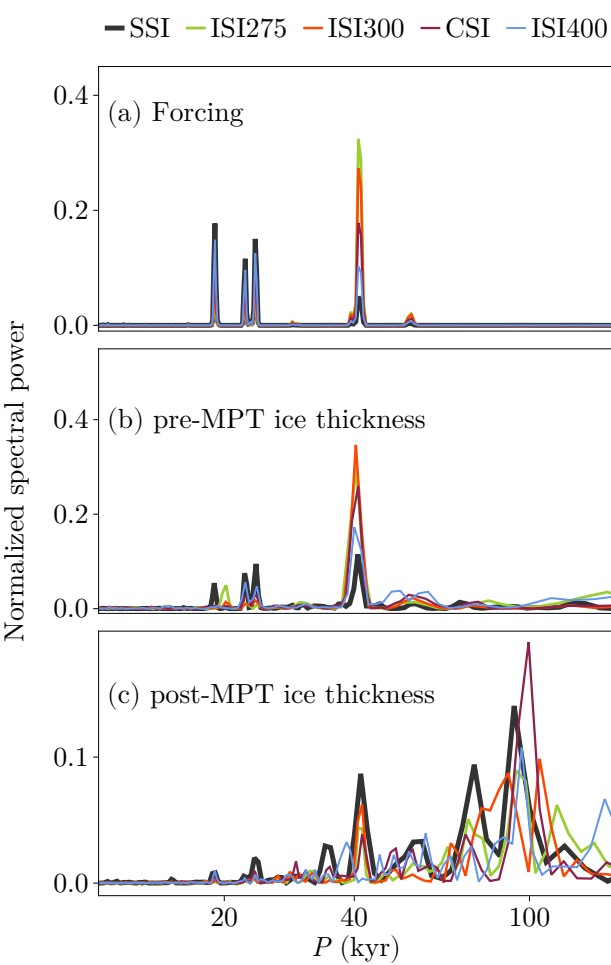

**Figure 12.** Normalized spectral power of (a) insolation forcing, (b) pre-MPT and (c) post-MPT ice thickness corresponding to each forcing in the INSOL experiment.



## 3.5 Exploring hypotheses beyond the regoliths

BASE results fundamentally rely in the dynamic change of the sediment layer thickness. Thus, a plausible question is: What
would happen to our results if we do not consider the sediment layer evolution? In other words, what if the regoliths formed
during the Pliocene were removed from the continents long before the MPT? To this end, we performed two experiments
deactivating the sediment layer dynamics: NOSEDIM-high and NOSEDIM-low (Table 2, Fig. 13). These experiments present
the same model setup as the AGING configuration in Sect. 2.7 from Pérez-Montero et al. (2024). The only difference is that we
started them at 3 Myr BP and applied two different basal sliding parameter values: $C_s = 6 \cdot 10^{-5}$ m yr$^{-1}$ Pa$^{-2}$ for NOSEDIM-
high (the same as BASE) and $C_s = 3 \cdot 10^{-8}$ m yr$^{-1}$ Pa$^{-2}$ for NOSEDIM-low (the same value as in Pérez-Montero et al., 2024).
As can be seen in Fig. 13, during NOSEDIM experiments, the model does not transition between pre- to post-MPT worlds.
Instead, the model remains in the dynamic regime determined by the imposed sliding intensity: 20-40-kyr world with high
sliding and 80-120-kyr world with low sliding. This fact was expected since there is no change in the system behavior that
alters the mass balance of the ice sheet. In this context, we will analyze the effect of introducing a cooling trend in the climate
and whether that trend is capable of altering the mass balance of the ice sheets sufficiently to provide a change in the system
periodicity when sliding is not interactive with basal sediment thickness.

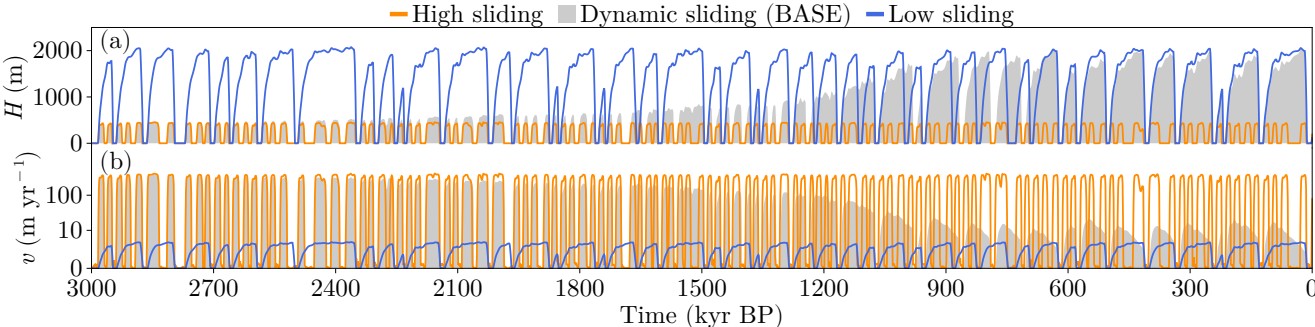

**Figure 13.** Time series of (a) ice thickness $H$ and (b) total velocity $v$ for BASE and NOSEDIM -high ($C_s = 6 \cdot 10^{-5}$ m yr$^{-1}$ Pa$^{-2}$, orange)
and -low ($C_s = 3 \cdot 10^{-8}$ m yr$^{-1}$ Pa$^{-2}$, blue).

### 3.5.1 The cooling effect of a trended carbon cycle

In Sect. 1 we noted that many studies contemplate the MPT as the consequence of a cooling trend in climate. Figure 14
shows the effect of adding this trend to NOSEDIM-low experiment. As can be seen, the trend in $C$ allow much higher accu-
mulation rates in the Early Pleistocene, producing higher amplitudes of $H$ relative to Late Pleistocene. This effect increases
monotonically with the trend applied until a threshold is reached (orange to red colors in Fig. 14a) and the ablation surpasses
accumulation, inhibiting glacial inception (Figure 14c). Again, we can see that the carbon cycle in PACCO only affects the
amplitude of GIV and has little influence on the periodicities except for the cases in which $CO_2$ is sufficiently (and probably



unrealistically) high at the Early Pleistocene. In fact, applying the same experiment to NOSEDIM-high (not shown) shows the
same behavior. At this point, one could ask: if the carbon cycle just amplifies the amplitude of GIV, did other climate-related
processes significantly change during the Pleistocene, allowing the observed ice-volume variation in proxies? We will answer
that question in the following section.

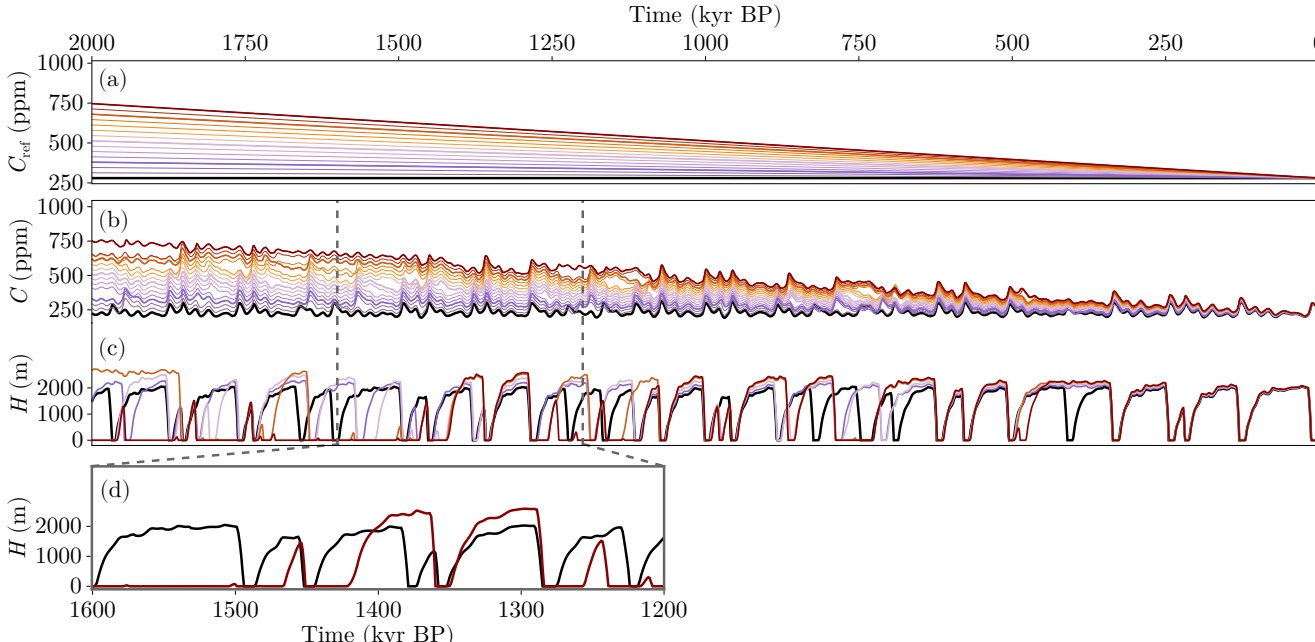

**Figure 14.** Time series of (colors) TREND-C experiment and (black) NOSEDIM-low experiment. (a) reference carbon dioxide $C_{\mathrm{ref}}$, (b)
carbon dioxide $C$, (c) ice thickness $H$ and (d) zoom of the range squared in (c) for the two extreme cases. Note that in (c) only the curves
highlighted in (a) and (b) are plotted.

### 3.5.2  What if the hydrologic cycle changes over time?

The classical interpretation of $\delta^{18}$O implies lower ice-sheet volumes during the Early Pleistocene. In the previous section we
have seen that in our results, the carbon cycle mainly just modifies the amplitude instead of facilitating any change in the
periodicities of the glacial cycles. It is, however, plausible that climate sensitivity (in its large sense) had changed during the
Pleistocene. Thus, here we wonder about the effects that, for instance, a variable hydrological cycle could have on the MPT. To
answer that question we performed an experiment applying a trend to the sensitivity of snowfall accumulation to temperature
$k_{\dot{s}}$ (Section A3, Eq. A6) to NOSEDIM-low (Figure 15). As can be seen, increasing $k_{\dot{s}}$ allows a change in the GIV regime
across the Pleistocene (Figure 15d). This change comes from the fact that if a higher accumulation sensitivity to temperature is
allowed, the ice sheet is more reactive to changes in its mass balance. Thus, at the Early Pleistocene, the ice sheet evolves more



strongly with insolation forcing for a certain level of $k_{\dot{s}}$. After several hundred of thousand of years, that sensitivity is reduced and the solution converges to NOSEDIM-low and hence to the Late Pleistocene simulated in BASE.

In summary, although imposing an ad-hoc decreasing trend in $CO_2$ does not produce an MPT in our model, we show that a progressive reduction of the sensitivity of precipitation to temperature does facilitate the shift in periodicities observed during the MPT. This indicates that perhaps the MPT could have been triggered or facilitated by another climate subsystem that influences the hydrological cycle sensitivity (e.g., precipitation pattern, Gildor and Tziperman, 2001)

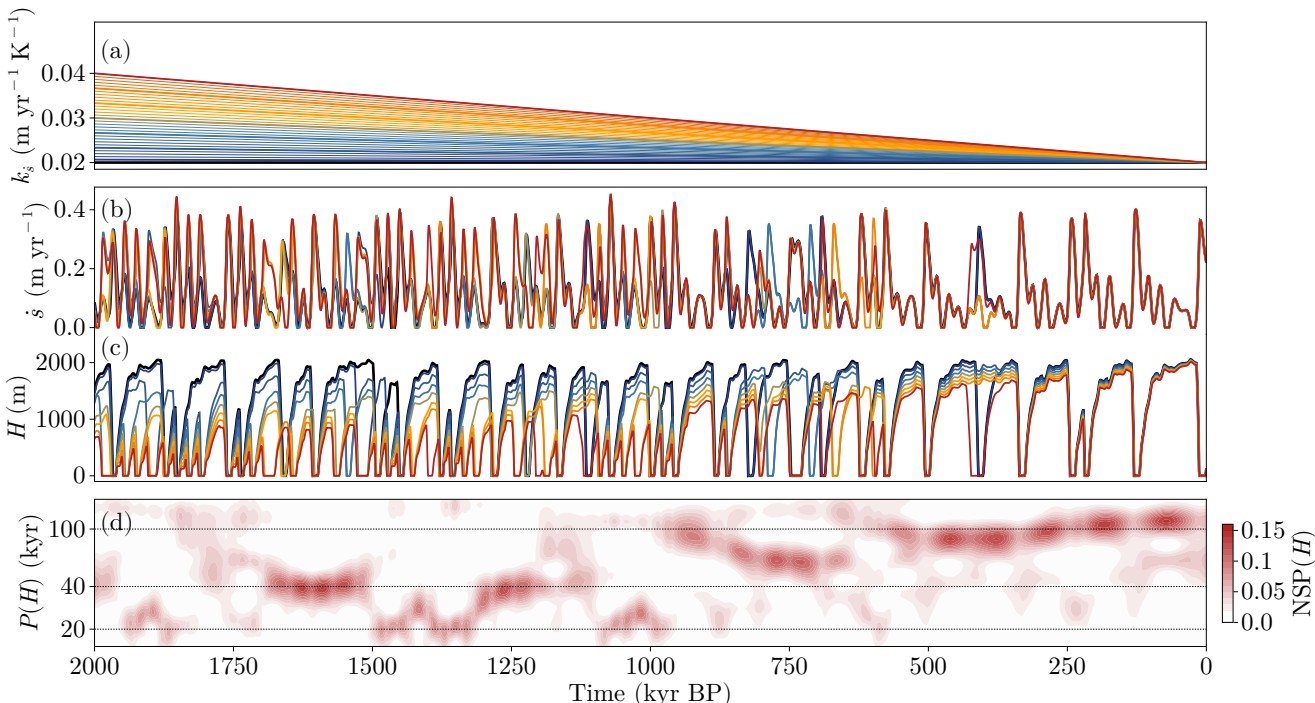

**Figure 15.** Time series of (colors) TREND-S experiment and (black) NOSEDIM-low experiment. (a) sensitivity of accumulation to temperature $k_{\dot{s}}$, (b) snowfall $\dot{s}$, (c) ice thickness $H$ and (d) wavelet transform of $H$ for the maximum decreasing trend (red) case. Note that in (b) and (c) only the curves highlighted in (a) are plotted.

## 4   Discussion

In this work we have simulated the entire Pleistocene with the spatially adimensional and conceptual climate-cryosphere model
PACCO (Pérez-Montero et al., 2024). To this end, we have added a new process to the original model that accounts for the existence of a dynamic and soft sediment layer beneath the Northern Hemisphere ice sheets. This hypothesis is based on the evidence of remnants of these layers over the Northern Hemisphere continents (Fig. 2, Pisias and Moore Jr, 1981; Berger and Jansen, 1994; Clark and Pollard, 1998; Clark et al., 2006; Ganopolski and Calov, 2011; Willeit et al., 2019; Carrillo et al., 2024).





The weak regolith substratum facilitates glacier basal sliding thus enhancing ice discharge. This mechanism is self-limited
since ice flow itself partially remove the sediments during glacial stages, thus reducing basal sliding and therefore ice flow. The
sediment is only partially regenerated during the comparatively short interglacials. PACCO captures these processes through the
representation of the hypothetical imbalance between sediment flux and denudation rate that produced the current distribution
of sediments along the continents (Fig. 2). This imbalance produces two different dynamic regimes along the Pleistocene. At
the early Pleistocene, high levels of sediments allow higher levels of ice discharge that facilitate a full deglaciation via an
increase in insolation due to precession and obliquity cycles (Fig. 5b). Subsequently, levels of sediments are lower and thus,
ice sheets grow bigger and persist longer. At this point, deglaciations are triggered by the manifestation of different nonlinear
processes, among which the albedo darkening facilitates simulating the right timing (Fig. 5c, Pérez-Montero et al., 2024).
Therefore, under this simple but physical formulation, the potential sliding of the Northern Hemisphere ice sheets, modulated
by the basal dynamics, plays a main role in triggering the MPT and produces the observed trend in climate proxies due to
the cooling effect of larger ice sheets in the Northern Hemisphere. The phase space of the sediment layer equation has been
tested for a wide range of the parameters involved in the SEDIM experiment. The imbalance between these two parameters
defines the rate at which sediments are removed beneath the ice sheets. Therefore, it determines the moment where the change
of dynamic regime is produced. There is no reason for this relationship to be linear and we do not exclude the existence of
better representations of the sediment quarrying and generation. However, Eqs. (4) and (5) are enough to show that a gradual,
interactive change in the dynamic regime can, in principle, alter the response of the climate system to insolation forcing along
the Pleistocene, in agreement with Legrain et al. (2023).

In addition, the role of $CO_2$ as the GIV amplifier has been explored. Our model reproduces GIV for constant levels of $C$
as long as the prescribed background level remains low enough. The CARBON experiment conclusions align with previous
work since it is the average value rather than its fluctuations that allows and amplifies GIV in PACCO. This finding implies
that in PACCO, the trend in $CO_2$ throughout the Pleistocene can be explained as a consequence of the increased ice-sheet size
due to slowing ice dynamics. However, this fact should not necessarily be interpreted as the definitive mechanism leading to
the atmospheric $CO_2$ trend shown by proxies and reconstructions during earlier epochs (Van de Wal et al., 2011; Willeit et al.,
2015).

We have also seen that the selection of the particular metric for insolation forcing facilitates the pre-MPT world to oscillate
at 40 kyr. Thus, one could see this as a simple forcing choice problem, but it could also hide the role of the Antarctic ice
sheet, the pole antisymmetry of precession cycles or the latitudinal insolation gradient (Raymo and Nisancioglu, 2003; Raymo
et al., 2006). Still, metric selection or uncertainty about the pre-MPT world does not change the post-MPT world and thus our
conclusions about the regolith hypothesis remain valid.

Finally, we tested the role of a cooling trend in climate by separately applying trends in the carbon cycle (Section 3.5.1,
TREND-C) and in snowfall accumulation (Section 3.5.2, TREND-S). Again, we have seen that a trend in PACCO's carbon
cycle only amplifies GIV but it can not produce a change in its frequency. However, as shown in Fig. 15, a trend in snowfall
is capable of producing a change of periodicity along the Pleistocene. In this case, we play directly with the mass balance
using snowfall sensitivity to temperature as another forcing. Where does this extra forcing comes from? We can not say which





is the physical mechanism behind because of the simplicity of its implementation. However, one could hypothesize possible
mechanisms that could imply such a change: In a warmer climate, the hydrological cycle is amplified, thus being more sensitive
to changes in short scale fluctuations of temperature. Then, if air temperature (governed in these timescales by solar radiation)
changes, the mass balance of the ice sheet can be affected more strongly. However, in a colder climate, the hydrological cycle
is weakened and less sensitive to solar fluctuations.

The present work shows that the regolith hypothesis can produce a quantitative match of the $CO_2$ and temperature proxies,
which were used to formulate the competing hypothesis of a cooling climate as the main trigger of the MPT. Although this does
not discard other mechanisms that could have cooled the climate, it shows that the regolith hypothesis mechanistically explains
the cooling and is coherent with proxy records, while being supported by the current sediment distribution at high northern
latitudes. In comparison, the cooling climate hypothesis suffers a lack of mechanistic explanation to date, as weathering was
likely low since the climate was relatively cold and dry. Additionally, no major change in tectonic or biological processes
happened over this time, potentially discarding them as carbon sinks. Of course, it is difficult to come to a final conclusion
without proxies with a more accurate time resolution covering the Early Pleistocene.

## 5   Conclusions

In summary, we tested the regolith hypothesis using a conceptual but physically-derived approach. The results show good
agreement with all proxy records covering this period. We conclude that the ice-sheet dynamic interaction with the regolith
layers could have played an important role in the change in the oscillatory regime during the MPT, in agreement with paleo
evidence of glacier erosion (Goldich, 1938; Wang et al., 1981; McKeague et al., 1983; Raymo et al., 1989; Setterholm and
Morey, 1995; Sosdian and Rosenthal, 2009; Rohling et al., 2014; Hodell and Channell, 2016). The sensitivity of the equation
describing this mechanism has been explored and we have found that the model captures the transition well for a wide range
of values. However, these values must be chosen based on a partial imbalance between the quarrying term and the contribution
from precipitation. This imbalance ensures correct timing according to the proxies.

As stated in Sect. 1, other models based on the carbon-cycle hypotheses produce a MPT in agreement with paleoclimatic
records. Here, we have shown that this is not a necessary condition to produce GIV in agreement with proxies. It rather
reproduces the variability of $CO_2$ thanks to the interplay between insolation, air temperature and ice sheets. Furthermore, our
model reproduces the observed trends in paleoclimate proxies as a consequence of the ice sheet size and not as the cause.
Thanks to its modularity (Pérez-Montero et al., 2024), PACCO could serve as a tool to test the mechanisms related to the
carbon cycle, as well as the role of the Antarctic ice sheet and even additional components of the climate system (as shown in
Sect. 3.5).

Finally, we have seen that the definition of insolation forcing is important for a good representation of the early Pleistocene,
as other studies have suggested (e.g. Leloup and Paillard, 2022). However, as far as the specific mechanism that triggered the
MPT is concerned, our results do not change with the particular insolation forcing applied.





**Table A1.** This table complements Table 1. Note that the parameters not referenced correspond to model calibration values.

| Parameter | Name | Reference | Value (range) | Units |
|---|---|---|---|---|
| $A_{\mathrm{thr}}$ | Ice-sheet size thermal anomaly | Paleoclimatic constraint | 20 | K |
| $B_{\mathrm{ref}}$ | Reference bedrock elevation | | 500 | m |
| $c_C$ | Climate sensitivity to $CO_2$ | | 0.65 | K m$^2$ W$^{-1}$ |
| $c_I$ | Climate sensitivity to insolation | | 0.09 | K m$^2$ W$^{-1}$ |
| $c_z$ | Climate sensitivity to ice surface elevation | | 0.007 | K m$^{-1}$ |
| $I_{\mathrm{ref}}$ | Reference insolation | Present-day anomaly | 480 | W m$^{-2}$ |
| $k_I$ | Ablation sensitivity to insolation | | 0.027 | yr$^{-1}$ W$^{-1}$ m$^{-1}$ |
| $k_{\dot{s}}$ | Snowfall sensitivity to temperature | | 0.02 | m yr$^{-1}$ K$^{-1}$ |
| $k_{T,C}$ | $CO_2$ sensitivity to temperature | | 5 | ppm K$^{-1}$ |
| $L_{\mathrm{ocn}}$ | Potential oceanic boundary | Margold et al. (2015) | $10^6$ | m |
| $L_{\mathrm{lb}}, L_{\mathrm{ub}}$ | Ice sheet horizontal scale constraints | Bates and Jackson (1987) | 200, 3000 | km |
| $T_{\mathrm{ref}}$ | Reference temperature | Present-day anomaly | 0 | °C |
| $T_{\mathrm{thr}}$ | Ablation threshold temperature | | -5 | °C |
| $\alpha_{\mathrm{l}}$ | Land albedo | Cuffey and Paterson (2010) | 0.9 | |
| $\alpha_{\mathrm{ni}}, \alpha_{\mathrm{oi}}$ | Albedo values for new and old ice | Willeit and Ganopolski (2018) | 0.9, 0.4 | |
| $\lambda$ | Ablation sensitivity to temperature | | 0.1 | m yr$^{-1}$ K$^{-1}$ |
| $\tau_\alpha$ | Albedo aging time scale | Willeit and Ganopolski (2018) | $50 \cdot 10^3$ | yr |
| $\tau_B$ | Bed relaxation time | Le Meur and Huybrechts (1996) | $5 \cdot 10^3$ | yr |
| $\tau_C$ | $CO_2$ relaxation time | | 10 | yr |
| $\tau_T$ | Temperature relaxation time | | 900 | yr |

*Code and data availability.* PACCO is available at https://github.com/sperezmont/Pacco.jl. The archived version of the code in this paper can be found at https://doi.org/10.5281/zenodo.14534680. The code to generate all the figures of the document and its archived version can be found at: https://github.com/sperezmont/Perez-Montero-etal_2025_CP and https://doi.org/10.5281/zenodo.14891899.

## Appendix A: Selected PACCO model equations

### A1  Regional air temperature

The evolution of air temperature is treated "regionally" as the interaction between insolation, carbon cycle and the presence of ice sheets in the Northern Hemisphere via

$$\frac{dT}{dt} = \frac{[T_{\mathrm{ref}} + c_I \cdot R_I + c_C \cdot R_C - c_z \cdot z] - T}{\tau_T}, \tag{A1}$$

where $c_I, c_C, c_Z$ are the climate sensitivities to insolation forcing ($R_I = I - I_{\mathrm{ref}}$), $CO_2$ radiative effect

$$R_C = 5.35 \cdot log\left(\frac{C}{280}\right), \tag{A2}$$

following Myhre et al. (1998) and ice-sheet surface elevation $z$ respectively. Note that the "ref" index indicates a reference value for $T$ and $I$. $\tau_T$ is the air thermal relaxation time for a reference state perturbation.





## A2 Carbon cycle

The carbon cycle in PACCO is represented by

$$\frac{dC}{dt} = \frac{[C_{\mathrm{ref}} + k_{T,C} \cdot (T - T_{\mathrm{ref}})] - C}{\tau_C}, \tag{A3}$$

where $k_{\mathrm{T,C}}$ is a parameter that converts the anomaly of temperature to a change in the carbon dioxide concentration. In this way, PACCO mimics the transient behavior of some physical processes that are not resolved (e.g., ocean biogeochemistry) because of its lack of spatial resolution.

## A3 Mass conservation

The ice-sheet evolution is given by the usual mass conservation equation (e.g., Benn et al., 2019) that accounts for snowfall ($\dot{s}$), ablation ($\dot{a}$) and ice discharge across the grounding line ($v \cdot H \cdot L_{\mathrm{ocn}}^{-1}$) via

$$\frac{dH}{dt} = \dot{s} - \dot{a} - v \cdot \frac{H}{L_{\mathrm{ocn}}}, \tag{A4}$$

with

$$q = v \cdot \frac{H}{L_{\mathrm{ocn}}}, \tag{A5}$$

$$\dot{s} = \dot{s}_{\mathrm{ref}} + k_{\dot{s}} \cdot (T - T_{\mathrm{ref}}) \tag{A6}$$

and

$$\dot{a} = k_I \cdot (1 - \alpha) \cdot (I - I_{\mathrm{ref}}) + \lambda \cdot (T - T_{\mathrm{thr}}), \tag{A7}$$

where $\dot{s}_{\mathrm{ref}}$ is the reference snowfall, $k_{\dot{s}}$ the snowfall sensitivity to temperature anomaly and $\alpha$ the time-evolving albedo of the system (Sect. 2.7 from Pérez-Montero et al., 2024). $k_I$ and $\lambda$ are the sensitivities of ablation to insolation and temperature respectively. Finally, $T_{\mathrm{thr}}$ is a model parameter that measures the point where we assume thermal ablation is representatively happening in the entire ice sheet.

## A4 Ice volume

Due to the lack of spatial dimensions in PACCO, the ice volume $V_{\mathrm{ice}}$ needs to be defined as a relation between the prognosed states in order to compare with paleoclimatic records. Thus, we defined $V_{\mathrm{ice}}$ (in meters of sea level equivalent, m SLE) as

$$V_{\mathrm{ice}} = -\frac{\rho_{\mathrm{ice}}}{\rho_{\mathrm{wtr}} \cdot S_{\mathrm{ocn}}} \cdot S(T, v_b) \cdot H, \tag{A8}$$

where $\rho_{\mathrm{wtr}}$ is water's density, $S_{\mathrm{ocn}}$ is the oceanic surface of the Earth and

$$S = S_1 + \frac{v_b}{\nu} \cdot \left[ \pi \cdot L_{\mathrm{ub}}^2 - S_1 \right], \tag{A9}$$



with

$$S_1 = \frac{T_{\text{ref}} - T}{A_{\text{th}}} \cdot \pi \cdot L^2. \tag{A10}$$

This is a parameterization of how the surface potentially glaciated ($\pi \cdot L^2$) is modified due to regional temperature anomalies ($\Delta T \cdot A_{\text{thr}}^{-1}$) and basal sliding of the ice sheet ($v_b \cdot \nu^{-1}$). Here $L$ is the characteristic horizontal scale of the ice sheet

$$L = c \cdot z^2, \tag{A11}$$

adapted from Verbitsky et al. (2018).

## Appendix B: Insolation metrics

Figure B1 shows the time series and wavelet transforms of each forcing applied in Sect. 3.4.

*Author contributions.* JAS and SPM conceived PACCO. SPM implemented PACCO, performed the analysis, created the figures and tables, and wrote the paper. JSJ improved the code efficiency and structure. DMP largely contributed to conceptualise the governing equations of ice-sheet thermodynamics. JAS, JSJ, DMP, MM, and AR provided extensive feedback on the analysis and the article.

*Competing interests.* At least one of the (co-)authors is a member of the editorial board of Climate of the Past.

*Financial support.* This research has been supported by the Spanish Ministry of Science and Innovation (project IceAge, grant no. PID2019-110714RA-100 and project CCRYTICAS, grant no. PID2022-142800OB-I00). AR received funding from the European Union (ERC, FOR-CLIMA, 101044247). JSJ is funded by the ClimTip project, which has received funding from the European Union's Horizon Europe research and innovation programme under grant agreement No. 101137601. This is ClimTip contribution 33. DMP is supported by the Fonds de la Recherche Scientifique - FNRS under Grant n° T.0234.24.





**Figure B1.** Insolation metrics compared. Note that the horizontal gray dashed line indicates the reference value employed for each metric (which is the value at present day).





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
