# Peer review of "Understanding the Mid-Pleistocene transition with a simple physical model"

_EGUsphere, 2025_

## Author Comment (AC1)

**Response to L. Lisiecki, RC1 ([https://doi.org/10.5194/egusphere-2025-2467-RC1](https://doi.org/10.5194/egusphere-2025-2467-RC1)) Climate of the Past, July 18, 2025**

Dear Lorraine Lisiecki,

Thank you for your comments, we here after respond point by point:

*This paper uses a recently published simple model, the Physical Adimensional Climate Cryosphere Model (Perez-Montero et al., 2025), to investigate several hypotheses about the Mid-Pleistocene Transition (MPT). They find that removal of North American regolith caused by repeated glaciations throughout the Pleistocene (i.e., the regolith hypothesis) is capable of reproducing the increase in ice volume amplitudes and switch to longer ~100-kyr cycle lengths in the mid-Pleistocene. The model also simulates increases in glacial-interglacial amplitudes (i.e., decreasing glacial minima) for CO2 and regional surface temperatures as a consequence of ice volume changes, which the manuscript argues demonstrates that a change in CO2 forcing is not necessary to drive these changes. The manuscript also includes many sensitivity tests to explore how the model responds to different parameter choices, insolation forcing, different constant CO2 levels, and other potential climate system changes in order to investigate mechanisms that are and are not associated with transitions in the model.*

*This manuscript is well written and thoroughly explores the dynamics of the model associated with MPT-like glacial cycle changes. The work is novel, well-executed and likely to be of interest to many paleoclimate researchers. However, there are a couple issues that should be addressed before its publication in Climate of the Past.*

*Major concerns*

*For comparing the model results to paleoclimate observations of the MPT, the manuscript (e.g., Figure 4) mainly relies on results from Bintanja & van de Wal (2008) and Berends et al (2021), both of which use forward inverse modeling to infer temperature and ice volume from benthic d18O. This is a concerning limitation of current manuscript because it relies on the accuracy of assumptions in these inverse models. While there are no direct observations of CO2 across and before the MPT, there are direct observations of regional temperatures to which the authors could compare their model results and there are estimates of ice volume that don't rely on the same inverse modeling assumptions. The authors should compare their model results to some of these more observationally based estimates, such as*

*NH extra-tropical SST records:*

*Clark, P. U., Shakun, J. D., Rosenthal, Y., Köhler, P., and Bartlein, P. J.: Global and regional temperature change over the last 4.5 million years, Science, 383, 884–890, 2024.*

*McClymont et al. , Earth Sci. Rev. 123, 173–193 (2013).*

*Lawrence et al, North Atlantic climate evolution through the Plio-Pleistocene climate transitions, EPSL, 300, 329–342, 2010.*

*Ice volume estimates:*

*Elderfield, et al (2012) Evolution of ocean temperature and ice volume through the mid-Pleistocene climate transition, Science, 337, 704–709, https://doi.org/10.1126/science.1221294.*

*Rohling et al (2014) Sea-level and deep-sea-temperature variability over the past 5.3 million years, Nature, 508, 477–482.*

*Clark et al (2025), Mean ocean temperature change and decomposition of the benthic d18O record over the past 4.5 million years Clim. Past, 21, 973–1000, https://doi.org/10.5194/cp-21-973-2025*

*In describing these comparisons, it would also be appropriate for the manuscript to clarify that regional SST records would be expected to systematically differ from temperature over the ice sheet because of different sensitivities to ice sheet height*

We thank the reviewer for these constructive comments and have taken them into account. We now show in Fig. RC1.1 some of the proxies suggested compared  with the related variables of PACCO. In addition, we attach a new version of Fig. 4 for the BASE experiment (Fig. RC1.2) that includes the comparison with the suggested proxies, where we splitted the last row in order to compare more clearly $V_{ice}$ and $\delta^{18}O$. Concerning the proxies:

- We added Herbert et al. (2016) instead of Lawrence et al. (2010) since their studies are similar and the data for the first one is publicly available but not for the second one (the first author of Lawrence et al (2010) seems to not be in academia anymore).
- Clark et al. (2025) provides a deconvolution of $\delta^{18}O$ in its thermal and sea water components, but not the relative sea level (ice volume), and it is not within our scope or area of expertise to make the conversion required for a direct comparison.
- We added $\delta^{18}O$ from Hodell et al. (2023) that was already included in Figure 1 from the manuscript.

[Figure]

**Figure RC1.1.** Comparison between the suggested proxy records (coloured lines) and our simulations (black). The new variables included are global mean surface temperature (GMT) and sea surface temperature (SST).

[Figure]

**Figure RC1.2.** New Figure 4 for BASE experiment.

*Because the new pre-MPT ice volume estimates of Clark et al (2025) differ substantially from most other study's reconstructions (and this model's output), I recommend that the authors add some discussion of this new study somewhere in the manuscript.*

As far as we know, Clark et al. (2025) have only published the deconvolution of the δ¹⁸O but not the ice-volume estimates, thus we can not compare the sea-level estimates. We agree that the conclusion of their deconvolution is interesting since it could yield pre-MPT and post-MPT ice volume changes of the same amplitude while conserving the 40 and 100 kyr periodicities. This is however controversial since it conflicts with the regolith hypothesis proposed in Clark and Pollard (1998). However, we think that other proxies such as the ice rafted detritus also need to be explained by this new hypothesis. We have therefore included a paragraph on this issue in the discussion section.

*More generally, the model seems to produce very small ice volume estimates right up until the MPT (e.g., see 1500-1200 kyr BP in Figure 4f). Do the authors have any comments about this aspect of the*

*model response? Is the model specifically producing estimates of Laurentide ice volume or all NH ice sheets?*

It is important to note that our ice-volume equation is based on a very basic assumption concerning ice-sheet geometry. Thus, our ice-volume estimates cannot be compared one to one with more sophisticated reconstructions of this magnitude. The reason behind the low values of ice volume before the MPT is related with the geometric assumption that relates ice thickness to the horizontal profile. This relationship helps with ice dynamics but strongly limits how much ice the ice sheet can store. The model represents the ice volume of the Northern Hemisphere as a whole, but neither of these ice sheets were purely symmetrical. Therefore, the comparison must be qualitative rather than quantitative. It is also important to say that in reality, the regoliths were spatially distributed and removed. Thus, some parts could accumulate more ice as the regoliths were removed sooner in some places than in others. Therefore providing bigger ice volumes than our equation. On the other hand, our simulated pre-MPT ice volume is indeed infraestimated compared to proxies. Please see our response to referee #2 (RC2), where we expand on the influences of the summer solstice insolation   forcing and the absence of the Antarctic ice sheet on this underestimated ice volume previous to the MPT.

*Minor comments*

- *Lines 205-207: The sentence at the bottom of page 16 is somewhat difficult to parse. One way to clarify it (I think) would be "This improvement ... trigger the MPT, whereas the decreasing trends ... occur as consequences of changes in the cryosphere in this model."*

We agree, thanks for the suggestion.

- *Lines 223-236: The authors show model results for different CO2 trends in Figure 14 and on line 228 summarize the results by saying the carbon cycle affects the amplitude of GIV but "has little influence on the periodicities." However, it is difficult to discern the periodicities of glacial cycles in Figure 14c, particularly because results from different model runs are overlain. The manuscript should provide more specific documentation of constant periodicity between the early and late Pleistocene in these simulations, perhaps by plotting NSP for a couple of the TREND-C experiments (e.g., similar to Fig. 15d).*

We agree, thanks for the nice suggestion.  You can see in Fig. RC1.3 that the frequencies are very similarly distributed between cases. The only change is related with the inception and the amplitude. Moreover, we think that this figure can be a good addition to the appendix of the paper.

[Figure]

**Figure RC1.3.** NSP as a function of time for NOSEDIM-low and 4 emblematic cases of the TREND-C experiments. Note that the associated H is plotted in grey as a reference. (a) Interactive carbon cycle and the trend applied in the rows below is (b) -0.05 ppm/kyr, (c) -0.12 ppm/kyr, (d) -0.20 ppm/kyr and (e) -0.23 ppm/kyr.

- *Line 266: Please specify which "two parameters" are referred to here.*

We meant fv and fp; we will clarify this in the new version of the manuscript.

- *Line 289: The sentence "We cannot say which is the physical mechanism behind because of the simplicity of its implementation" is unclear. Perhaps one or more words is missing after "behind."*

We will change this to "We cannot identify the physical mechanism underlying the change in snowfall sensitivity to temperature because of the simplicity of its implementation." in the new version of the manuscript.

- *The citation for Perez-Montero et al. (2025) should be updated to the final, published version instead of the pre-print.*

That is right, the article was published after submitting this manuscript. We will update the citation.

Sincerely,
Sergio Pérez-Montero et al.

---

## Author Comment (AC2)

**Response to RC2 (https://doi.org/10.5194/egusphere-2025-2467-RC2) Climate of the Past, Sept 01, 2025**

Dear reviewer,

Thank you for your comments, we here after respond point by point:

*This study investigates the MPT using the conceptual but physically based model PACCO (Physical Adimensional Climate Cryosphere Model). The model incorporates insolation forcing, $CO_2$, ice dynamics, and the evolution of regolith layers. Modelling simulations suggest that the progressive removal of regolith slowed ice-sheet flow, allowing larger ice sheets that match the timing and amplitude of the MPT seen in proxy records. Simulations including CO2 trend but constant sediment layers are not conclusive for simulating MPT. Overall, the findings support the regolith hypothesis as a plausible mechanism for the MPT, but also that hydrological sensitivity may have contributed as well.*

*The approach addresses a crucial question in paleoclimate research: the trigger of the MPT. The choice of model to test the regolith hypothesis, related to the spatial distribution of regolith through time beneath the ice sheet is unconventional, given that the study relies on a 0-D physical model. The study effectively leverages the model flexibility by testing several hypothesis and conducting sensitivity tests. However, the results of the model require more thorough explanation to support the conclusions, in my opinion.*

*The manuscript is generally well written and structured. However, the discussion section lacks sufficient contextualisation with respect to previous studies, does not provide a critical assessment of the authors work, and suffers from a lack of clear structure.*

*Main comments:*

*1. The regolith hypothesis is inherently geographically-based and has already been tested using 2D models (e.g., Willeit et al. 2019). In particular, the spatial evolution of regolith patches beneath the ice sheets is expected to play a critical role in their stability. For this reason, investigating the hypothesis with a spatially adimensional model is, in my view, not an obvious or straightforward approach. At a minimum, a section describing the implications and nuances of this approach choice in the discussion section is necessary. However, the manuscript currently lacks any explicit acknowledgment of model limitations or a critical appraisal of the robustness of the results.*

We understand this concern, but must stress that conceptual models are still useful for a better understanding of the long-term climate evolution, as still reflected in recent literature on the MPT (e.g. Verbitsky et al., 2018, Leloup and Paillard, 2022, Verbitsky and Crucifix, 2023, Koepnick and Tziperman, 2023, Ganopolski, 2024 among others). Of course, an adimensional model will fail in capturing the potential consequences of a spatially-heterogeneous regolith evolution on the overall basal dynamics of the Northern Hemisphere ice sheets. The same applies to the spatial distribution of accumulation and ablation. However, in spite of our adimensional approximation, our results show a successful characterisation of the MPT periodicity change facilitated by the inclusion of the effects of the regolith on basal ice dynamics. Furthermore, all processes are simulated in a physically consistent (not empirical or conceptual) manner. On the other hand, we acknowledge that others

(e.g. Willeit et al., 2019) have already nicely addressed the implication of the regolith in 2D models. But, because of the absence of an explicit module for the interactions between sediments and basal ice dynamics, the assumptions for regolith evolution remain simple even at 2D: a simple progressive shift from a full regolith coverage at 3Ma (with an ad-hoc enhanced sliding factor of 5) to the present-day sediment mask. This study was indeed very inspiring and we are currently working on implementing an explicit basal sediment module in our 3D ice-sheet model, but this is out of the scope of the current manuscript. Nevertheless, we will acknowledge the limitations of our approach inherent from its adimensional character and will accordingly discuss the implications.

*2. The BASE simulation, which serves as the reference for sensitivity tests and comparisons across different hypotheses, does not adequately reproduce the amplitude of the 40 kyr world (Fig. 4f), but only approximately a quarter of its amplitude. As a result, the climate state preceding the MPT is not well simulated in the model. Indeed, the too low amplitude of the climatic 40 ka cycles likely preserve a large quantity of regolith until the start of the MPT.  How does this limitation affect the simulation of the MPT under the various hypotheses tested?*

It is true that in the BASE simulation pre-MPT sea-level variations are underestimated compared to other studies or to the $\delta^{18}O$ signal. This is the case for the original proxies we used (Bintanja and van der Wal, 2008; Berends et al., 2021b). However, following a suggestion by referee 1 (see RC1 for more details), we have adapted the figure to include a more proxy-based reconstruction (Elderfield et al. 2012). Figure RC1.2 clearly shows an improvement in the comparison.

In addition, compared with the $\delta^{18}O$ signal, this underestimation varies from 3-1.1 Ma. It is minor and only present for a few glacial maxima between 3-1.8 Ma, and of the order of a half for 1.8-1.1 Ma. However, this is indeed a limitation of our study (although not necessarily a limitation of our modelling approach). There are two main potential causes for the misrepresentation of the pre-MPT cycles already addressed in the existent literature for the MPT:

1. The $65^{o}$ boreal summer solstice insolation is known to overestimate the local effects of precession and therefore produces an enhanced response around a 23 kyr periodicity (Leloup and Paillard  2022).
2. The Antarctic ice sheet contributed substantially to sea-level variations in the pre-MPT 40-kyr world. As stated in Raymo et al. (2006), because the Earth's orbital precession is out of phase between hemispheres, precession-related changes in ice volume in each hemisphere would cancel out while the-inphase obliquity-induced changes in Antarctica and in NH ice volume would add up.  Of course this cannot be captured by PACCO since it only simulates the Northern Hemisphere ice sheets.

Elucidating the exact cause of the obliquity vs precession pre-MPT sea-level amplitude issue is not directly addressed in our paper. However, underestimating the pre-MPT amplitude is indeed a limitation of our study and we will accordingly acknowledge this in the new version of the manuscript by expanding the discussion section and reflecting this discussion.

On the other hand, this limitation does not affect the post-MPT simulation in any manner or the other hypotheses tested here. First, the experiments described in section 3.5 are carried out by

unplugging the interactive regolith dynamics and second, as shown in section 3.4, when our model captures better the 40-kyr world it also simulates a very good post-MPT period.

*A related question is the choice of insolation forcing. Using an alternative insolation metric (CST or ISI instead of SSI) allows the model to better capture the 40 kyr amplitude, yet in doing so the MPT itself is simulated much earlier than in the paleoclimate records, which is likely due to the larger quantity of regolith removed during the 40 ka cycles. The choice of insolation metric in the BASE model (SSI) versus alternatives such as CSI or ISI is not clearly justified and likely has a larger impact on the study results than the authors suggest.*

First, SSI was chosen for historical reasons and for preserving Milankovitch's initial hypothesis (as for example, in Ganopolski, 2024's Generalized Milankovitch Theory). SSI is indeed usually employed in conceptual modeling (Paillard, 1998; Parrenin and Paillard, 2003; Leloup and Paillard, 2022; Ganopolski, 2024) and it is cheaper to compute than CSI and ISI.

Second, the sediment module was calibrated against paleorecords using the SSI forcing. The regolith-related parameters could be slightly re-tuned for CSI and the MPT would be simulated at the right timing (see figure RC2.1 below). However, we decided to keep the same regolith parameters as in BASE when exploring other insolation metrics for clarity. In this way, the effects of varying the different insolation forcings are visible on the periodicity-change of the system under the exact same set of parameters as BASE and thus only reflect the effects of insolation. Nevertheless, we agree that we should include a sentence to justify the choice of SSI and will do so in the new version of the manuscript.

[Figure]

**RC2.1.** BASE experiment using CSI forcing (BASE-CSI). Note that we only changed $f_p$ to 1e-7 (-29% of the original value in CSI experiment) and $f_v$ to 1.15e-5 (-62% of the original value in CSI experiment). Note that, in light of the Referee's comment, we will also include this figure in Appendix B.

*3. This study is in line with a broader set of modeling efforts aimed at simulating the MPT over the past decade. Several of these studies, however, have reached conclusions that differ from those presented here. I strongly encourage the authors to expand the discussion by explicitly comparing their results with previous work. A concrete example, though not the only relevant one, is the recent publication by Scherrenberg et al. (2025) in this journal. It would be valuable to discuss whether the differences arise from the type of model employed, the assumptions underlying the hypotheses, or the specific formulation of the model.*

The main difference with Scherrenberg et al. (2025) is that we only force the model using insolation. However Scherrenberg et al (2025)'s forcing index includes the CO2 signal. In our opinion this actually forces the orbital-scale response during the Pleistocene in their model. In our case, the orbital-scale response emerges from the model physics. This agrees with studies with models more comprehensive than ours suggesting that CO2 variations are not critical to produce glacial-interglacial cycles (e.g. Abe-Ouchi et al 2013; Willeit et al 2019). Finally, our conclusions do not differ much from those of recent studies. For example, Verbitsky et al. (2018) stated that the MPT is the product of a change in the balance of feedbacks in the system, Willeit et al. (2019) showed in a 3D model that regolith can produce the MPT and Ganopolski (2025) summarized the

main processes that govern GIV which include the change in dynamic regime and also the importance of the ice-sheet size.

The new version of the manuscript will include an expanded discussion to compare with previous results. We appreciate the suggestion.

*4. I am quite surprised by the absence of effect of a decreasing trend in CO2, with concentrations starting as high as 700 (!) ppm down to the late Pleistocene values, on the triggering of the MPT. In another hand, the model is able to switch from the 40 to 100 kyr world only by changing hydrological component (i.e. accumulation sensitivity to temperature). Both of these results are quite different from the rest of the literature and would deserve further discussion and comparison with other studies in the discussion section.*

There is a simulated effect of starting at such high CO2 concentrations in our model (see the related minor comment below). However, the CO2 trend is not enough to trigger MPT in our case, and it plays a lesser role on the ice evolution than for example in Scherrenberg et al. (2015) or in Willeit et al. (2019). In Scherrenberg et al (2025), a strong influence of a decreasing CO2 trend on sea-level amplitude is expected because, by construction, the reconstructed CO2 trend is used to build the climate index utilized to force the ice-sheet model. As for the comparison to Willeit et al. (2019) we both have an active (not imposed) carbon cycle, thus the reason for the lower influence of the decreasing CO2 in our case is likely our adimensional assumption: It is conceivable that under high CO2 atmospheric concentrations a global model such as CLIMBER-2 will inhibit glacial inception, even under cold orbits, in continental areas where ablation remains high, so that those glacial maxima will show a relatively low amplitude. In our case, however, due to the lack of spatial dimensions, if those same cold orbits allow for inception, the ice sheet will grow according to the mean conditions without reflecting any spatial heterogeneity. We will acknowledge this fact in the new version of the manuscript and expand the discussion accordingly with particular emphasis on the comparison with previous studies.

On the other hand, concerning section 3.5 (hydrological component) we will also expand the discussion on the potential implications of such new findings. It is worth noting here that the search for MPT hypothesis beyond the regoliths and the decreasing CO2 was motivated by the new deconvolution of $\delta^{18}O$ done in Clark et al. (2025). Even though sea-level is not explicitly reconstructed in that study, by qualitatively exploring their new temperature reconstruction, one might infer that the associated pre-MPT ice volume amplitude has been classically underestimated. If so, the regolith hypothesis might have to face a conundrum: How could the pre-MPT ice-sheet maxima be as big as the post-MPT in a 40-kyrs world? Section 3.5 might shed light on such a question if a new sea-level reconstruction motivating it is finally presented.

Therefore, we will again expand the discussion by including this latter context in the new version of the manuscript.

*Title: Consider using "Exploring" instead of "Understanding" to reflect a less ambitious and more accurate scope.*

We agree and will change the title accordingly.

*Abstract: In my opinion, too much space is given to the introduction, with too little devoted to summarizing the study methods and results. The long sentence that carries all results (line 9-11) should be split into at least two shorter sentences. The last sentence is overly descriptive and does not capture the main conclusion.*

We thank you for your nice suggestion and we will modify the abstract accordingly.

*Line 21: The citation of Chalk 2017 is not the most appropriate reference to introduce the concept of the MPT, go for a more historical one.*

We agree. We will use Shackleton (1987), Lisiecki and Raymo (2005) and Clark et al. (2006) as references to the MPT.

*Line 23: This sentence should be moved up, as the described feature is an inherent part of the MPT itself, not an additional aspect.*

We agree and modify the manuscript accordingly.

*Line 31: References are need to support the existence of the regolith. Currently, all cited works relate only to the second part of the sentence.*

We agree. The new references will be:

- Setterholm and Moorey (1995)
- Clark et al. (1999)
- Goss and Rooney (2023)

*Line 24: The paragraph is quite dense. it would be better to start a new paragraph line 31 to improve clarity.*

We thank you for the suggestion and we will implement it in the new version of the manuscript.

*Line 50: The sentence is quite reductive of the work done, as the authors also test the $CO_2$ decrease hypothesis.*

We agree and we will change it accordingly.

*Line 51 and following: I would avoid describing the paper section by section. The current structure is standard and the descriptions here are not helpful because too vague.*

Agreed. We will delete those lines.

*Line 114: It should be mentioned that while there is a change in amplitude and frequency, the model does not reproduce the amplitude of 40 kyr cycles. It seems the BASE model transitions from a quasi-stable climate directly to 100 kyr cycles, which is not equivalent to reproducing the MPT. This is*

*nuanced by the fact that the 41 kyr cycles are better captured using other insolation metrics. However, why were these other insolation not applied across all simulations, instead of SSI?*

Please see above our previous responses concerning the choice of insolation forcing. Applying other insolation forcings (e.g. CSI) to all sections does not alter the main findings of the study and we believe it would deviate reader's attention from the well-structured and compartmentalized outline of the current progressive experimental setup. See also figure RC2.1.

*Line 115 and following: At this stage, it may be premature to draw such a conclusion. In my view, it is the constant sediment simulation that provides stronger evidence for that.*

We agree. We will change those lines accordingly.

*Figure 8: How is defined the Tmpt value ? I guess it is an arbitrary choice, but it needs to be justified in the text.*

$t_{MPT}$ is defined in Equation 7 and justified in text as: "since we empirically observed a threshold in the oscillatory regime around that value of sediment thickness".

*Figure 9: It would be interesting to do the same plot but with sea level value (as in fig. 4f) to see how it vary compared to the sea level curves.*

We plotted H since it is one of the main prognostic variables of the model, and we believe that because the manuscript is already quite lengthy and has 15 figures (plus the the former figure of Appendix B and the new figures RC2.1 and RC.1.3 to be added in the Appendix as well), adding new plots is not desirable.

*Line 200: Interesting observation. However, the authors note that with CSI and ISIn, 40 kyr cycles are larger and thus remove sediments earlier. This relates to a key criticism of the regolith hypothesis: much of the regolith would already be removed during the initial 40 kyr cycles. Could the good match between modeled regolith removal and MPT timing be due to the BASE experiment's failure to produce realistic 40 kyr cycles compared to data?*

We think this is not the case. The good match of BASE with the proxies is intrinsic to the change in dynamic regime, but it is not related by any means with the periodicities before the MPT. The CSI and ISIn experiments remove more sediments because we kept $f_v$ and $f_p$ as in BASE. This is important and the reason why we carried out the SEDIM experiment. Our formulation of $H_{sed}$ evolution is simple and thus, the election of $f_v$ and $f_p$ is key to the correct timing of the MPT. Therefore, our good match with BASE is not due to SSI forcing but due to the fact that the change in the dynamic behavior of the ice sheet is produced at the right time. Please see also our response to main comment #2 and figure RC2.1.

*Line 205-207: I disagree with this conclusion. The change in frequency and amplitude occurs very early compared to the "real" timing of the MPT.*

The conclusion is that the change in dynamic regime due to the removal of regolith is the trigger of the MPT. We were not saying that the timing was good, but the mechanism is. See also our answer to

the referee's main comment #2 where we show a CSI simulation with tuned regolith parameters allowing not only the right pre-MPT frequency but also a good MPT timing.

*Line 224: The absence of an impact on the trend in frequency and amplitude is quite surprising, especially in light of previous modeling studies on the MPT (e.g. Scherrenberg et al., 2025, CP; Willeit et al., 2019; Science Advances).*

The influence of the decreasing $CO_2$ is indeed limited when comparing to other studies (see main comment #3) but it does exist (see new figure RC1.3): The runs with an initial $CO_2$ above 600 ppm show a tendency to inhibit many glacial maxima between 2 and 1.5 Ma. Note, for example in the red curve of figure 14, the presence of only a few short (precession-induced) and low-amplitude glacials during that period. For the second part of the runs (1 - 0 Ma), it can be seen that higher $CO_2$ translates into slightly thicker ice sheets (because of the enhanced accumulation associated with a warmer atmosphere), with little frequency impact. This effect is progressively attenuated from the initial $CO_2$ range of approximately 500 - 250 ppm until the end of the simulations.

*Line 234: This section and the results here are quite surprising, and to my knowledge, the first modelling study that proposes the hydrological cycle as a trigger of the MPT.*

Thank you for this comment. We will try to stress out the novelty of the section and will expand the discussion accordingly (see main comment #4).

*Line 244: The sentence here, will describing accurately the results of this study, sounds very surprising.  See main comment 4 for more details.*

(see our response to main comment #4)

*Line 284 and after. The paragraph should be reworked, as is it poorly written: e.g.  "Of course, it is difficult to come to a final conclusion without proxies with a more accurate time resolution covering the Early Pleistocene."*

We have rephrased it: "Having a paleorecord with more accurate time resolution covering the Early Pleistocene would help to shed light on the mechanisms explored in modelling studies".

*Typos:*

*Line 8: "sediment layers of sediments above the continents"*

*Ligne 105 : "beyon the regoliths"*

*Ligne 225 : allows*

*Ligne 255 : removes*

We appreciate your careful observation of these typos and we will correct them in the new version of the article.

*References:*

*Willeit, M., Ganopolski, A., Calov, R., & Brovkin, V. (2019). Mid-Pleistocene transition in glacial cycles explained by declining CO2 and regolith removal. Science Advances, 5(4), eaav7337.*

*Scherrenberg, M. D., Berends, C. J., & van de Wal, R. S. (2025). CO 2 and summer insolation as drivers for the Mid-Pleistocene Transition. Climate of the Past, 21(6), 1061-1077.*

Sincerely,
Sergio Pérez-Montero et al.

**References**

Clark, P. U., Alley, R. B., & Pollard, D. (1999). Northern Hemisphere ice-sheet influences on global climate change. *Science*, *286*(5442), 1104-1111.

Ganopolski, A. (2024). Toward generalized Milankovitch theory (GMT). *Climate of the Past*, *20*(1), 151-185.

Goss, G. A., & Rooney, A. D. (2023). Variations in Mid-Pleistocene glacial cycles: New insights from osmium isotopes. *Quaternary Science Reviews*, *321*, 108357.

Leloup, G., & Paillard, D. (2022). Influence of the choice of insolation forcing on the results of a conceptual glacial cycle model. *Climate of the Past*, *18*(3), 547-558.

Paillard, D. (1998). The timing of Pleistocene glaciations from a simple multiple-state climate model. *Nature*, *391*(6665), 378-381.

Parrenin, F., & Paillard, D. (2003). Amplitude and phase of glacial cycles from a conceptual model. *Earth and Planetary Science Letters*, *214*(1-2), 243-250.

Raymo, M. E., & Nisancioglu, K. H. (2003). The 41 kyr world: Milankovitch's other unsolved mystery. *Paleoceanography*, *18*(1).

Raymo, M. E., Lisiecki, L. E., & Nisancioglu, K. H. (2006). Plio-Pleistocene ice volume, Antarctic climate, and the global δ18O record. *Science*, *313*(5786), 492-495.

Scherrenberg, M. D., Berends, C. J., & van de Wal, R. S. (2025). CO 2 and summer insolation as drivers for the Mid-Pleistocene Transition. *Climate of the Past*, *21*(6), 1061-1077.

Setterholm, D. R., & Morey, G. B. (1995). *An extensive pre-Cretaceous weathering profile in east-central and southwestern Minnesota* (No. 1989). US Government Printing Office.

Verbitsky, M. Y., Crucifix, M., & Volobuev, D. M. (2018). A theory of Pleistocene glacial rhythmicity. *Earth System Dynamics*, *9*(3), 1025-1043.